# EvaLearn: Quantifying the Learning Capability and Efficiency of LLMs via Sequential Problem Solving

**Shihan Dou**[1]*, **Ming Zhang**[1]*, **Chenhao Huang**[1], **Jiayi Chen**[1], **Feng Chen**[2,3],
**Shichun Liu**[1], **Yan Liu**[2], **Chenxiao Liu**[2], **Cheng Zhong**[2], **Zongzhang Zhang**[3], **Tao Gui**[1,4]†,
**Chao Xin**[2], **Chengzhi Wei**[2], **Lin Yan**[2], **Qi Zhang**[1,5]†, **Xuanjing Huang**[1,5]†

[1]Fudan University  [2]ByteDance Seed  [3]Nanjing University  [4]Shanghai Innovation Institute
[5]Shanghai Collaborative Innovation Center of Intelligent Visual Computing
{shdou24, mingzhang23}@m.fudan.edu.cn
{tgui, qz, xjhuang}@fudan.edu.cn

## Abstract

We introduce **EvaLearn**, a pioneering benchmark designed to evaluate large language models (LLMs) on their learning capability and efficiency in challenging tasks, a critical, yet underexplored aspect of model potential. EvaLearn contains 648 challenging problems across six task types, grouped into 182 sequences, each sequence dedicated to one task type. Diverging from most existing benchmarks that evaluate models in parallel, EvaLearn requires models to solve problems sequentially, allowing them to leverage the experience gained from previous solutions. EvaLearn provides five comprehensive automated metrics to evaluate models and quantify their learning capability and efficiency. We extensively benchmark nine frontier models and observe varied performance profiles: some models, such as Claude-3.7-sonnet, start with moderate initial performance but exhibit strong learning ability, while some models struggle to benefit from experience and may even show negative transfer. Moreover, we investigate model performance under two learning settings and find that instance-level rubrics and teacher-model feedback further facilitate model learning. Importantly, we observe that current LLMs with stronger static abilities do not show a clear advantage in learning capability across all tasks, highlighting that EvaLearn evaluates a new dimension of model performance. We hope EvaLearn provides a novel evaluation perspective for assessing LLM potential and understanding the gap between models and human capabilities, promoting the development of deeper and more dynamic evaluation approaches. All datasets, the automatic evaluation framework, and the results studied in this paper are available at `https://github.com/ByteDance-Seed/EvaLearn`.

## 1 Introduction

Large language models (LLMs) have advanced rapidly in recent years, driving remarkable progress in a wide range of applications [90; 9; 65; 26; 57; 4]. Rigorous evaluation of these models is essential for understanding their current capabilities, identifying areas for improvement, and guiding the development of more advanced LLMs [10; 39; 27; 25; 102].

Although numerous benchmarks have been proposed to assess various aspects of model performance, the overwhelming majority rely on a parallel evaluation paradigm [93; 71; 43; 7; 47; 40; 33; 85]. In this setting, models are tested on independent and identically distributed (i.i.d.) samples, and their overall performance is reported as aggregate metrics. However, such benchmarks primarily

---

* Equal contribution.
† Corresponding author.

39th Conference on Neural Information Processing Systems (NeurIPS 2025).

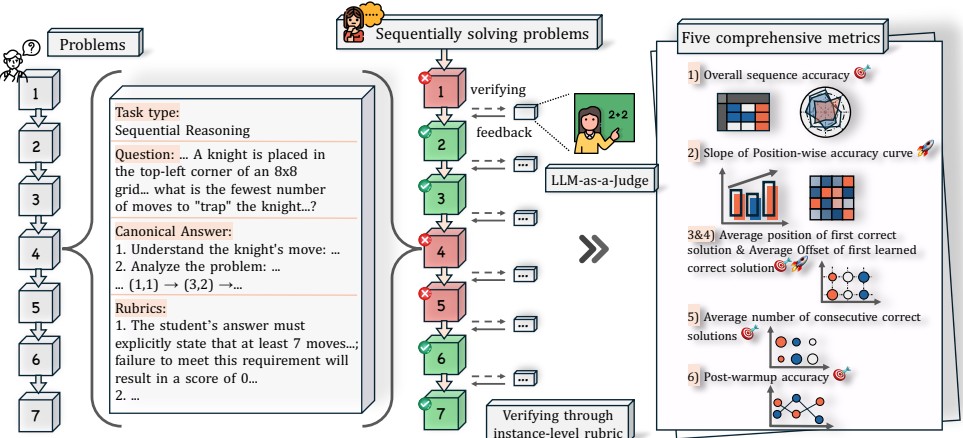

Figure 1: Overview of EvaLearn. Instead of parallel evaluation, EvaLearn requires models to solve problems sequentially, thereby systematically evaluating the LLM's learning ability and efficiency.

measure static abilities, while overlooking an equally important dimension: a model's capability to learn and adapt within a specific task (i.e., **learning capability**), as well as the speed at which this learning occurs (i.e., **learning efficiency**). These dynamic learning abilities are fundamental indicators of human learning potential and intelligence [6; 20; 79; 36], yet remain largely unexplored in the evaluation of LLMs—mainly because the prevailing parallel evaluation paradigm is inherently unable to capture such learning dynamics.

To address this gap, we introduce EvaLearn, a challenging benchmark designed to systematically quantify the learning capability and efficiency of LLMs through a novel sequential evaluation paradigm. Considering the scarcity of carefully categorized and challenging related problems in existing benchmarks, we construct 648 challenging problems from scratch. These problems are organized into 182 sequences, with each sequence containing seven problems from the same task type and spanning six distinct task categories. Models are required to sequentially solve each problem within a sequence, with each solution automatically evaluated using a combination of instance-level rubrics and an LLM-as-a-judge [102; 69] framework. EvaLearn is designed to assess these two core aspects by testing whether models can leverage experience gained from solving previous problems to improve their performance on subsequent ones. Moreover, EvaLearn includes a suite of five metrics to comprehensively quantify the learning capability and efficiency of a model. Importantly, these metrics are decoupled from the specific learning methods employed (i.e., how models utilize prior experience from earlier problems), highlighting the extensibility and flexibility of EvaLearn.

We conducted a comprehensive study with nine frontier LLMs on EvaLearn. We summarize several key findings: **(a)** Models exhibit diverse learning capabilities across different task types. They are generally more adept at leveraging prior experience to solve tasks involving mathematical and logical reasoning, while tasks such as summarization tend to rely more on knowledge acquired during pre-training and the model's inherent abilities. Moreover, learning efficiency also varies significantly across tasks. **(b)** Thinking-based LLMs typically outperform non-thinking-based LLMs in both learning capability and learning efficiency. They are better able to utilize experience to solve new problems and also show greater learning stability, being more likely to solve multiple problems consecutively within a sequence. **(c)** Feedback learning, which enables models to solve problems with the help of feedback from previous solutions and rubric-based evaluations, significantly enhances both the learning capability and efficiency of models. Moreover, it is often more effective than demonstration learning, which simply provides previous problems and canonical solutions as context. **(d)** Indicators of learning capability and learning efficiency should be considered jointly. These metrics together provide a comprehensive assessment of a model's learning potential. Moreover, they are not strongly correlated with static model capability. Even LLMs with higher static performance do not demonstrate a clear advantage in learning capability across all tasks.

More insightful findings are presented in Section 3 and Appendix G. We hope EvaLearn offers a new perspective on assessing the learning potential of LLMs, which is a key indicator of human-like intelligence. EvaLearn has a significant potential to foster a deeper understanding of model

capabilities within the community, promote the development of more effective learning methods, and serve as a pioneering step toward more dynamic and realistic evaluation paradigms.

## 2 EvaLearn

In this section, we first describe dataset composition and task types. We also provide an example to clearly illustrate the data structure. Section 2.2 then presents the automated evaluation procedure based on instance-level rubrics. Section 2.3 introduces five metrics we designed to quantify the learning capability and efficiency of LLMs. We also present the data annotation process in Appendix E.

### 2.1 Datasets

EvaLearn is designed to evaluate the LLM's learning capability and efficiency. It contains 648 challenging problems, all carefully constructed from scratch and organized into 182 sequences. Each sequence contains seven problems under the same task category. Problems within a sequence are interrelated and collectively challenge the model's learning potential. Figure 1 provides an overview of EvaLearn. Each problem within a sequence includes a question and a canonical answer. Since most of these challenging questions cannot be reliably evaluated using pre-defined rules, each problem is also accompanied by a human-written rubric that defines the criteria for assessing the correctness of model responses (details on the evaluation process are provided in Section 2.2).

In contrast to most existing benchmarks that evaluate models in a parallel setting, EvaLearn requires models to solve problems sequentially within each sequence. This sequential setup is specifically designed to evaluate a model's ability to accumulate experience and leverage feedback from previous problems, as reflected in its performance on subsequent problems within the same sequence.

**Task types.** EvaLearn comprises six distinct task types, with each sequence belonging to one of these categories. These task types include: **(1) Summarization (Sum)** that evaluates whether models can improve the accuracy and coverage of summaries by leveraging prior experience; **(2) Classification (Cla)** that assesses a model's ability to enhance its categorization skills from solving a series of classification problems; **(3) Extraction (Ex)** that measures whether models can progressively improve the accuracy and completeness of key information extraction; **(4) Logical Reasoning (LR)** that tests whether models can learn from previous errors and improve logic reasoning ability; **(5) Mathematical Reasoning (MR)** that examines whether models can quickly master mathematical problem-solving methods by utilizing feedback from earlier problems; **(6) Sequential Reasoning (SR)** that evaluates whether models can enhance their ability to solve sequence-based problems by learning from historical experience, including clarifying event steps and reasoning logic.

Table 1 presents the statistics of EvaLearn. Across all tasks, the average word counts per question is 315.45. Solving these problems requires models to possess strong and diverse capabilities. The results and analysis for various models on EvaLearn are provided in Section 3.

**An example of sequence.** We use an example sequence to showcase the relationships among problems within a sequence, as illustrated in Figure 23, which involves solving a $4 \times 4$ sliding puzzle. In this case, the puzzle consists of 16 tiles, one of which is a blank space (denoted by '>'), while the remaining tiles are filled with different symbols. By moving the blank space and swapping it with adjacent tiles, the goal is to restore the puzzle from a given initial state to a target configuration through a series of logical moves. This task is designed to assess the model's logical reasoning ability, and all problems within this sequence are centered around the sliding puzzles and logical reasoning. Figure 24 presents another problem from the same sequence, which differs mainly in grid size, the initial state of the puzzle, and symbol representation.

These problems are related and together assess the model's capability for logical reasoning. Humans are able to gain experience and improve their performance by solving a series of such problems [38; 76]. By requiring models to solve these problems sequentially within a sequence, EvaLearn evaluates whether models can similarly learn from experience, thereby measuring their learning efficiency.

### 2.2 Automatic Evaluation

Most of the challenging problems in EvaLearn cannot be reliably evaluated using rule-based verifiers, as many questions may have answers that are difficult to verify with pre-defined rules or may allow for

Table 1: Statistics of EvaLearn.

| Task Type | # Problems | # Sequences | Problem reuse rate | Average words per question | Average words per canonical answer |
|---|---|---|---|---|---|
| Summarization | 60 | 16 | 1.87 | 959.97 | 220.79 |
| Classification | 48 | 13 | 1.90 | 203.18 | 149.31 |
| Extraction | 60 | 17 | 1.98 | 674.89 | 78.60 |
| Logical Reasoning | 360 | 102 | 1.98 | 227.49 | 81.98 |
| Mathematical Reasoning | 60 | 17 | 1.98 | 137.03 | 336.76 |
| Sequential Reasoning | 60 | 17 | 1.98 | 141.38 | 334.30 |
| Overall | 648 | 182 | 1.966 | 315.45 | 146.05 |

multiple correct answers. To address this, we employ an instance-level rubric combined with LLM-as-a-judge [25] to assess whether model outputs satisfy the corresponding rubrics. The evaluation prompt is shown in Figure 18, which includes three demonstrations to ensure that the judging model follows the instructions. In all experiments, we use GPT-4o as the judging model.

We conduct an additional experiment to validate the effectiveness of our automatic evaluation process. Specifically, for each problem in EvaLearn, we randomly select three models from a model list (as shown in Table 3) to generate answers, which are then evaluated using our evaluation framework. Human annotators further verify whether the judging model can successfully assess the correctness of the model outputs through the instance-level rubric. Results show that the evaluation accuracy exceeds 95% for all tasks, demonstrating the high reliability of our evaluation framework that combines instance-level rubrics with LLM-as-a-judge, consistent with findings from previous studies [69; 91].

## 2.3 Evaluation Metrics

In EvaLearn, models are required to sequentially solve all problems within a sequence. All evaluation metrics are computed based on the model's correctness across these problems. Notably, the choice of learning method, such as learning from demonstrations (i.e., in-context learning) [8; 53] or learning from feedback [66], is decoupled from the evaluation metrics. This design ensures that EvaLearn remains extensible and flexible, supporting a wide range of learning strategies.

Let $N = 182$ denote the number of sequences and $M = 7$ the number of problems in each sequence in EvaLearn. Let $y_{n,m} \in \{0, 1\}$ indicate whether the $m$-th problem in the $n$-th sequence is answered correctly (1 for correct, 0 for incorrect). We define the following five metrics to comprehensively evaluate the learning potential of models in sequential problem-solving:

**(1) Overall sequence accuracy** (Acc). This metric reflects the model's overall performance in sequential problem-solving and serves as an indirect indicator of its learning capability. It is computed as the average accuracy across all problems and all sequences:

$$\text{Acc} = \frac{1}{NM} \sum_{n=1}^{N} \sum_{m=1}^{M} y_{n,m} \tag{1}$$

Higher values indicate better overall performance. We further compute the **position-wise accuracy curve**, i.e., the accuracy at each problem position $m$ across all sequences:

$$\text{Acc}_m = \frac{1}{N} \sum_{n=1}^{N} y_{n,m} \tag{2}$$

Based on this curve, we define a helpful metric, i.e., the slope of the fitted accuracy curve, to capture dynamic learning behaviors.

**(2) Slope of fitted accuracy curve** ($k$): This metric measures the model's learning speed across all sequences by fitting a straight line to the position-wise accuracy curve using least squares regression [28]. It quantifies how quickly the model's accuracy improves as it progresses through sequences.

Specifically, let $\text{Acc}_m$ denote the average accuracy at position $m$ across all sequences. The fitted line is given by $y = kx + b$, where $x$ is the problem position and $k$ is the estimated slope obtained by minimizing the sum of squared errors:

$$k = \arg\min_{k,b} \sum_{m=1}^{M} \left( \text{Acc}_m - (km + b) \right)^2 . \tag{3}$$

A higher $k$ indicates greater learning efficiency and a faster rate of improvement.

**(3) Average position of first correct solution** ($P_{\text{first}}$). This metric measures the average position in the sequence where the model achieves its first correct solution, indicating its initial ability, learning capability, and efficiency. For each sequence $n$, let $p_n$ denote the position of the first correct solution (or $M + 1$ if none is correct). The metric is calculated as:

$$P_{\text{first}} = \frac{1}{N} \sum_{n=1}^{N} p_n \tag{4}$$

Lower values indicate that the model can achieve its first correct solution earlier in the sequence.

**(4) Average offset of first learned correct solution** ($P_{\text{offset}}$). This metric measures how quickly a model begins to learn within a sequence while discounting pre-existing knowledge. For each sequence $n$, let $i_0^{(n)}$ be the position of the first problem answered incorrectly under zero-shot (parallel) solving; if all problems are correct in zero-shot, set $i_0^{(n)} = M+1$. Let $A_{n,i}^{\text{zero}}, A_{n,i}^{\text{feedback}} \in \{0, 1\}$ denote correctness at position $i$ under zero-shot and under the sequential solving setting, respectively. We define the per-sequence offset to the first learned correct solution as:

$$d_n = \min \left\{ i - i_0^{(n)} \;\middle|\; i \geq i_0^{(n)},\; A_{n,i}^{\text{feedback}} = 1,\; A_{n,i}^{\text{zero}} = 0 \right\}, \tag{5}$$

with the convention that $d_n = M+1 - i_0^{(n)}$ if no such $i$ exists, and $d_n = 0$ when $i_0^{(n)} = M+1$. The metric is computed by averaging over sequences:

$$P_{\text{offset}} = \frac{1}{N} \sum_{n=1}^{N} d_n. \tag{6}$$

Lower values indicate that the model begins to learn earlier after excluding problems solved due to pre-existing knowledge.

**(5) Average number of consecutive correct solutions** ($N_{\text{consec}}$). This metric reflects the model's capability to leverage experience to consistently solve problems within a sequence, resulting in fewer errors. It indirectly indicates how effectively and stably the model learns. For each sequence $n$, we compute the longest run of consecutive correct solutions, then average across all sequences:

$$N_{\text{consec}} = \frac{1}{N} \sum_{n=1}^{N} \max_{1 \leq a \leq b \leq M} \{b - a + 1 : y_{n,a} = y_{n,a+1} = \cdots = y_{n,b} = 1\} \tag{7}$$

Higher values indicate greater consistency, while lower values suggest intermittent errors and less stable learning.

**(6) Post-warmup accuracy** ($\text{Acc}_{\text{pw}}$-K). This metric reflects model performance after an initial "warmup" phase, i.e., after some experience has been accumulated. For each sequence, we exclude the first $K$ problems and compute the average accuracy on the remaining problems:

$$\text{Acc}_{\text{pw}}\text{-K} = \frac{1}{N(M - K)} \sum_{n=1}^{N} \sum_{m=K+1}^{M} y_{n,m} \tag{8}$$

Higher values indicate better adaptation and learning after the warmup.

These metrics provide a comprehensive and nuanced evaluation of dynamic learning behaviors of models in sequential problem-solving, fully capturing learning capability and efficiency. In the following sections, we use these metrics to thoroughly analyze the learning potential of various LLMs.

## 3 Benchmarking LLMs with EvaLearn

In this section, we compare two problem-solving paradigms, including parallel solving and sequential solving, to investigate the learning capability and efficiency of LLMs.

### 3.1 Setup

**Parallel Solving.** This paradigm includes two settings: **(I) Zero-shot**: Models solve each problem independently, without access to any experience from previous problems. This setting aligns with

Table 2: Comparison of accuracy between zero-shot (parallel solving) and feedback Learning (sequential solving). Values in parentheses indicate the difference between the two methods. Full task names for the abbreviations can be found in Section 2.1. Blue-colored cells indicate the best performance for each task category under few-shot and feedback learning settings, respectively. All values are shown as %.

| Model | Paradigm | Sum | Cla | Ex | LR | MR | SR | Overall |
|---|---|---|---|---|---|---|---|---|
| **Non-thinking-based** | | | | | | | | |
| DeepSeek-V3 | Zero-shot | 81.7 | 72.9 | 45.0 | 25.6 | 71.7 | 60.0 | 43.5 |
| | Feedback Learning | 76.8 (-4.9) | 74.7 (+1.8) | 36.1 (-8.9) | 24.9 (-0.7) | 74.8 (+3.1) | 54.6 (-5.4) | 41.5 (-2.0) |
| Claude-3.7-Sonnet | Zero-shot | 76.7 | 64.6 | 48.3 | 8.1 | 48.3 | 33.3 | 28.4 |
| | Feedback Learning | 75.0 (-1.7) | 75.8 (**+11.2**) | 46.2 (-2.1) | 15.5 (**+7.4**) | 63.9 (+15.6) | 49.6 (**+16.3**) | 35.6 (**+7.2**) |
| GPT-4o | Zero-shot | 81.7 | 70.8 | 45.0 | 15.6 | 43.3 | 31.7 | 32.6 |
| | Feedback Learning | 78.6 (-3.1) | 73.6 (+2.8) | 48.7 (**+3.7**) | 12.9 (-2.7) | 61.3 (**+18.0**) | 32.2 (+0.5) | 32.7 (+0.1) |
| Doubao-1.5-Pro | Zero-shot | 71.7 | 70.8 | 45.0 | 10.3 | 61.7 | 41.7 | 31.3 |
| | Feedback Learning | 71.4 (-0.3) | 68.1 (-2.7) | 42.9 (-2.1) | 10.8 (+0.5) | 71.4 (+9.7) | 36.1 (-5.6) | 31.2 (-0.1) |
| Qwen2.5-32b-Instruct | Zero-shot | 66.7 | 60.4 | 31.7 | 6.9 | 46.7 | 21.7 | 23.8 |
| | Feedback Learning | 59.8 (-6.9) | 62.6 (+2.2) | 30.3 (-1.4) | 9.8 (+2.9) | 49.6 (+2.9) | 30.3 (+8.6) | 25.5 (+1.7) |
| **Thinking-based** | | | | | | | | |
| OpenAI-o3-mini | Zero-shot | 65.0 | 64.6 | 48.3 | 45.8 | 73.3 | 73.3 | 54.3 |
| | Feedback Learning | 75.9 (**+10.9**) | 73.6 (+9.0) | 47.1 (-1.2) | 59.9 (**+14.1**) | 80.7 (+7.4) | 78.2 (+4.9) | 64.8 (**+10.5**) |
| Doubao-1.5-Thinking-Pro | Zero-shot | 85.0 | 79.2 | 55.0 | 42.5 | 85.0 | 73.3 | 57.1 |
| | Feedback Learning | 82.1 (-2.9) | 70.3 (-8.9) | 52.9 (-2.1) | 39.4 (-3.1) | 77.3 (-7.7) | 55.5 (-17.8) | 51.6 (-5.5) |
| DeepSeek-R1 | Zero-shot | 86.7 | 89.6 | 48.3 | 41.7 | 78.3 | 66.7 | 55.7 |
| | Feedback Learning | 89.3 (+2.6) | 79.1 (-10.5) | 48.7 (+0.4) | 29.6 (-12.1) | 74.8 (-3.5) | 51.3 (-15.4) | 46.4 (-9.3) |
| Claude-3.7-Sonnet-Thinking | Zero-shot | 86.7 | 66.7 | 43.3 | 12.5 | 46.7 | 31.7 | 31.2 |
| | Feedback Learning | 78.6 (-8.1) | 80.2 (**+13.5**) | 48.7 (**+5.4**) | 18.8 (+6.3) | 58.0 (**+11.3**) | 46.2 (**+14.5**) | 37.4 (+6.2) |
| Gemini-2.5-Pro | Zero-shot | 85.0 | 72.9 | 55.0 | 62.5 | 90.0 | 76.7 | 68.5 |
| | Feedback Learning | 89.3 (+4.3) | 75.8 (+2.9) | 52.1 (-2.9) | 59.9 (-2.6) | 85.7 (-4.3) | 79.8 (+3.1) | 67.2 (-1.3) |
| Gemini-2.5-Flash | Zero-shot | 81.7 | 66.7 | 38.3 | 16.1 | 70.0 | 51.7 | 36.3 |
| | Feedback Learning | 67.9 (-13.8) | 59.3 (-7.4) | 42.0 (+3.7) | 18.3 (+2.2) | 81.5 (+11.5) | 58.8 (+7.1) | 37.5 (+1.2) |

the evaluation approach used in most existing benchmarks, assessing a model's inherent ability to solve challenging problems without any learning opportunity. The system prompt is shown in Figure 19. **(II) Few-shot**: For each problem, we provide three demonstrations from the same task as examples (i.e., 3-shot), offering models guidance on output format and problem-solving approach. The demonstrations are identical for all problems within each task type. The system prompt is shown in Figure 20.

**Sequential Solving.** To investigate models' ability to learn from experience, we utilize two sequential learning paradigms: **(I) Demonstration learning**: Models are provided with all previous problems and their corresponding canonical answers from the same sequence before solving the current problem, similar to in-context learning. The system prompt is shown in Figure 21. **(II) Feedback learning**: When solving the current problem, models receive as context all previous problems, their solutions, and detailed feedback on their own prior solutions, as assessed by a judge using instance-level rubrics. This setting evaluates whether models can leverage their previous experience to improve on subsequent problems. The system prompt is shown in Figure 22.

In all experiments, we evaluate nine frontier LLMs, including both thinking-based and non-thinking-based models: Claude-3.7-Sonnet [4], Claude-3.7-Sonnet-Thinking [4], DeepSeek-R1 [26], DeepSeek-V3 [45], Doubao-1.5-Pro [9], Doubao-1.5-Thinking-Pro [65], OpenAI-o3-mini [57], GPT-4o [55], and Qwen2.5-32b-Instruct [90]. See Appendix F for details on these models.

### 3.2 RQ I: Can LLMs learn a task by engaging with a sequence of problems?

> **Finding 1:** LLMs exhibit varying abilities to learn from problem sequences, with differences observed across both models and task types. Moreover, most models also demonstrate better performance after a warm-up phase.

Table 2 summarizes the differences in overall accuracy between the feedback learning and zero-shot paradigms. We observe that five models benefit from sequence learning, while four models experience slight declines, indicating that some models can effectively leverage prior experience to solve problems within the same task. Thinking-based models, in particular, exhibit more pronounced performance shifts, with OpenAI-o3-mini achieving the highest overall improvement (+10.5%).

Moreover, most LLMs show improvement on mathematical reasoning and classification tasks when learning from experience. For example, GPT-4o and Claude-3.7-Sonnet-Thinking achieve gains of

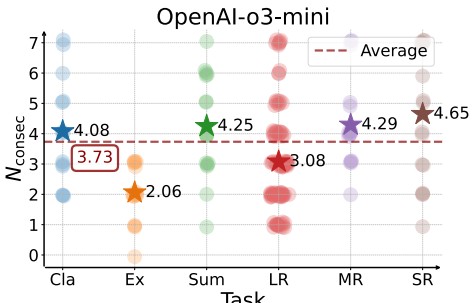 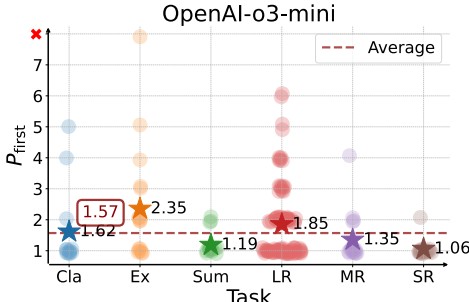

Figure 2: **(Left)** Average number of consecutive correct solutions ($N_{\text{consec}}$). **(Right)** Average position of the first correct solution ($P_{\text{first}}$). Results are shown for OpenAI-o3-mini, with each node representing a sequence.

18.0% and 13.5% in mathematical reasoning and classification, respectively. This improvement likely results from these tasks having clear solution steps, enabling feedback to pinpoint reasoning errors and help models effectively learn specific problem-solving strategies.

In contrast, most models experience performance declines on the summarization task; for instance, 7 out of 9 models perform worse after feedback in summarization. This may be because the summarization task rely more heavily on knowledge and instruction-following abilities acquired during pre-training, and additional experience may sometimes interfere with the model's ability to solve the current problem.

We also analyze models' Post-Warmup Accuracy ($\text{Acc}_{\text{pw}}$-K), with results for Claude-3.7-Sonnet-Thinking shown in the left side of Figure 3 and additional models presented in Appendix G.6. We observe that most models achieve higher accuracy in the later stages of the sequence, particularly among thinking-based models. This further demonstrates that models can use early problems as practice, leveraging the experience gained to solve subsequent, related problems more effectively.

> **Finding 2:** Learning stability varies significantly across different tasks and models. For certain tasks, such as summarization, current models are more adept at leveraging their inherent knowledge to solve problems rather than drawing on experience gained from previous problems.

We further investigate the learning stability of LLMs by analyzing the average number of consecutive correct solutions ($N_{\text{consec}}$). The left side of Figure 2 presents the results for o3-mini, with additional results for other models provided in Appendix G.5. Each colored node in the figure represents a sequence in the dataset, and nodes positioned higher indicate that the model solves more problems consecutively within that sequence.

From the results, we first observe that the average number of consecutive correct solutions varies notably across tasks. For most models, logical reasoning tasks present greater challenges for maintaining long streaks of correct solutions. For example, Claude-3.7-Sonnet and GPT-4o fail entirely or manage to solve only one problem consecutively in over half of the sequences. However, there are substantial differences between thinking-based and non-thinking-based models. For instance, while Doubao-1.5-Pro struggles to maintain consecutive correct solutions in logical reasoning, Doubao-1.5-Thinking-Pro exhibits much greater learning stability. Similar patterns appear in other model pairs, such as DeepSeek-V3 vs DeepSeek-R1 and GPT-4o vs o3-mini, where the thinking-based variants generally achieve higher average numbers of consecutive correct solutions. For example, o3-mini achieves an average maximum streak of 3.42 consecutive correct solutions per sequence, compared to only 2.58 for GPT-4o. Combining these results with Table 2, we conclude that thinking-based models not only benefit more from prior experience to improve performance, but also maintain higher stability and are more likely to solve multiple related problems in succession.

Interestingly, in the summarization task, models tend to have relatively high $N_{\text{consec}}$ values. However, these same models show a marked decline in $\text{Acc}_{\text{pw-}K}$, indicating that their performance on later problems in the sequence is worse than on earlier ones. This suggests that, for such tasks, the capability to solve multiple problems consecutively may rely more on the model's inherent knowledge and static ability, rather than on learning from experience within the sequence. This observation further supports the conclusion in Findings 1.

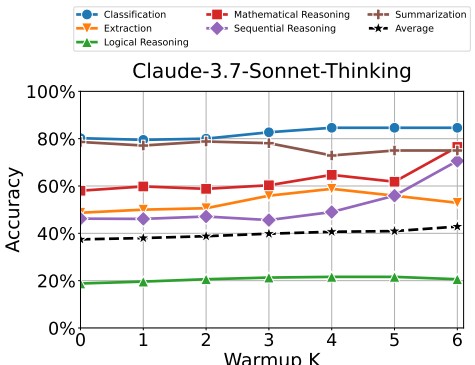 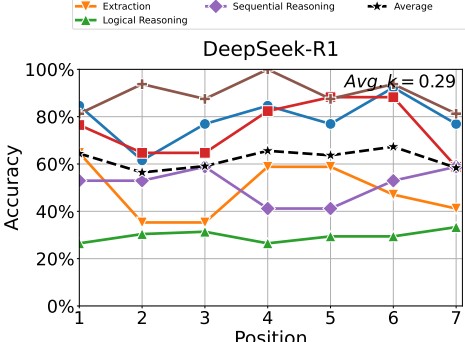

Figure 3: **(Left)** Post-warmup accuracy ($\text{Acc}_{\text{pw}}$-K) results of Claude-3.7-Sonnet-Thinking. **(Right)** Position-wise accuracy curve and its slope $k$ of DeepSeek-R1.

Overall, these metrics allow us to comprehensively assess the learning capability and stability of LLMs, providing deeper insight into their potential to learn from experience within a task.

> **Finding 3:** Learning capability provides a new perspective for evaluating models, independent of their static performance, and reveals their underlying learning potential.

Results in Table 2 show that strong performance in parallel solving does not necessarily imply strong learning capability. For example, although DeepSeek-R1 outperforms Claude-3.7-Sonnet in the parallel setting, it experiences a 9% drop in the sequential solving paradigm, while Claude-3.7-Sonnet achieves a 7.2% gain. In the logic reasoning task, Claude-3.7-Sonnet-Thinking does not outperform DeepSeek-R1 or Doubao-1.5-Thinking-Pro in the zero-shot setting, yet it demonstrates stronger learning capability by substantially improving its performance through experience. o3-mini, on the other hand, exhibits the best performance in this task under both zero-shot and feedback learning settings, showing remarkable learning capability by leveraging feedback to achieve a 14.1 percentage point improvement. Another example is DeepSeek-R1 in the math reasoning task. Although DeepSeek-R1 achieves higher zero-shot performance than o3-mini, it fails to learn from experience as effectively as o3-mini and instead suffers a decline in performance.

Moreover, models with similar static abilities may exhibit markedly different learning capabilities. For instance, Claude-3.7-Sonnet-Thinking and GPT-4o both achieve 31.7% accuracy in the sequential reasoning task under the zero-shot paradigm, yet their learning capabilities differ significantly: the former improves by 14.5 percentage points, while the latter only improves by 0.5 points. However, GPT-4o demonstrates stronger learning capability than Claude-3.7-Sonnet-Thinking in the math reasoning task. These results indicate that learning capability also varies across different tasks.

Taken together, these observations suggest that learning capability is an important and distinct metric for model evaluation, independent of static performance and varying across different tasks. This dimension of model assessment deserves further attention and systematic investigation by the research community. In Research Question III (§ 3.4), we further analyze model performance under various learning paradigms, providing additional evidence that learning capability is not solely determined by static ability.

### 3.3 RQ II: How efficient are LLMs at learning from a sequence of problems?

> **Finding 4:** Learning efficiency differs markedly across models and task types. On average, most non-thinking-based models improve more rapidly with experience, while thinking-based models tend to achieve more stable gains.

The right side of Figure 3 presents the position-wise accuracy curve for DeepSeek-R1, with corresponding curves for other models shown in Appendix G.7. The heatmap in Figure 7 summarizes the slope ($k$) of these curves, directly indicating model learning efficiency.

We observe substantial differences in learning efficiency across models. For instance, Claude-3.7-Sonnet achieves the highest overall learning efficiency, with $k = 2.08$. On average, most non-thinking-based models exhibit steeper slopes in their position-wise accuracy curves compared to

thinking-based models, as shown in Figure 7 (e.g., Claude-3.7-Sonnet, DeepSeek-V3, and GPT-4o). This may be because non-thinking-based models often start from a lower baseline, allowing them to quickly capitalize on "low-hanging fruit" as they accumulate experience.

In contrast, learning in thinking-based models is more stable. For example, as shown in Figure 16, o3-mini's position-wise accuracy curve shows a steady upward trend, even when its initial accuracy is already high. Non-thinking-based models often display larger fluctuations between adjacent positions, such as GPT-4o, DeepSeek-V3, and Doubao-1.5-Pro. This suggests that thinking-based models can better leverage information from previous problems, including feedback from the judging model, to reason more effectively about subsequent solutions, resulting in more stable performance gains. Moreover, most models display positive learning efficiency in math reasoning tasks. However, sequential reasoning tasks remain particularly challenging, with widespread performance declines; only Claude-3.7-Sonnet-Thinking shows meaningful positive learning on these sequences ($k = 1.89$). This indicates that even in reasoning tasks with clear solution paths and feedback, learning efficiency can vary significantly depending on the specific task, highlighting that it depends on multiple factors.

Overall, these results suggest that static model capability, task type, and reasoning approach all influence learning efficiency. Non-thinking-based models tend to improve performance more rapidly through experience, while thinking-based models achieve more stable improvements. Case studies in Appendix G.1 further illustrate how thinking-based models can utilize prior experience to solve new problems.

### 3.4 RQ III: Do different learning methods lead to differences in performance?

> **Finding 5:** Different solving approaches significantly affect model performance. Models can acquire experience from demonstrations, and feedback further enhances their learning. Moreover, learning capability is not strongly correlated with a model's inherent static ability.

Table 5 and Figure 6 present model performance across four solving methods. We observe that, for most models, demonstration learning in the sequential setting generally yields better results than few-shot parallel solving. For instance, Claude-3.7-Sonnet-Thinking achieves consistent performance improvements across five tasks compared to the few-shot setting. The key distinction is that demonstration learning allows models to access all previous problems and canonical solutions within the sequence, enabling them to learn from prior experience.

Furthermore, Figures 14, 15, and 9 compare feedback learning and demonstration learning in terms of post-warmup accuracy ($\text{Acc}_{\text{pw-}K}$) and the slope $k$, with additional results provided in Appendix G. Results show that, for most models, feedback learning achieves higher average overall accuracy and greater learning efficiency than demonstration learning. For example, DeepSeek-V3 and Doubao-1.5-Thinking-Pro show the ability to learn from feedback provided by the judging model and apply this experience to subsequent problems across five tasks.

Notably, we find that each LLM exhibits strong learning capability on certain tasks, but no model can consistently achieve stable learning improvements across all tasks. Even the state-of-the-art LLMs, as widely recognized in recent studies [77; 62; 68; 30; 78], such as o3-mini and Claude-3.7-Sonnet-Thinking, do not demonstrate a clear advantage in learning capability and efficiency across all tasks. Moreover, for every task, there are always some models that can learn from prior experience and improve their performance. For example, o3-mini exhibits strong learning capability in all tasks except sequential reasoning, whereas Claude-3.7-Sonnet can improve its performance on this task by leveraging experience from feedback.

These results indicate that each model has its own strengths, and that learning ability is not strongly correlated with static performance. Learning capability and efficiency thus provide a valuable new perspective for assessing model performance and understanding the gap between current models and human abilities. We also present a case study in Appendix G.1 to further illustrate how models learn from prior experience.

> **Finding 6:** The average position of the first correct solution $P_{\text{first}}$ and the average offset of the first learned correct solution $P_{\text{offset}}$ vary across models and tasks, providing important insights into model potential. All metrics in EvaLearn capture different aspects of a model's learning ability and efficiency, collectively revealing the model's learning potential.

We analyze the average position of the first correct solution $P_{\text{first}}$, which measures how quickly models achieve initial success within a sequence. The right side of Figure 2 presents $P_{\text{first}}$ for o3-mini, with results for other models provided in Appendix G.4. We observe that o3-mini and Doubao-1.5-Thinking-Pro achieve the best performance, with average positions of 1.57 and 1.80, respectively. These two models also achieve the highest overall accuracy in the feedback learning setting. Although $P_{\text{first}}$ is closely related to static model capability (since stronger models are more likely to solve problems earlier), notable exceptions exist. For example, DeepSeek-V3 achieves the same $P_{\text{first}}$ value as o3-mini on the mathematical reasoning task, yet its overall accuracy is 5.9% lower, indicating that o3-mini can better improve its problem-solving ability through experience.

We also find substantial variation in model performance across different tasks. For example, most models struggle to solve their first problem early in logical reasoning sequences, with only o3-mini and Doubao-1.5-Thinking-Pro consistently achieving early success. Additionally, comparing demonstration learning and feedback learning settings, as shown in Figures 11 and 10, seven out of nine models can solve a problem earlier when provided with feedback from the judging model. This indicates that, compared to simply providing canonical answers, feedback is more effective in facilitating model learning and mastery of a task. For instance, in the logical reasoning task, o3-mini sometimes fails to solve any problems in a sequence when using only demonstrations, but with feedback learning, it can solve at least one problem in every sequence.

Beyond $P_{\text{first}}$, we also analyze the average offset of the first learned correct solution $P_{\text{offset}}$, as shown in Table 4 . On average, frontier LLMs attain their first learned success relatively late in a sequence—typically around three to four additional positions after the first zero-shot miss, such as Qwen2.5-32b-Instruct, DeepSeek-R1, and DeepSeek-V3. By contrast, o3-mini tends to reach this point earlier on average. Clear task effects emerge. Models generally learn earlier on classification and extraction than on logical reasoning. For models such as GPT-4o and Qwen2.5-32b-Instruct, the first learned correct solution typically appears earlier in classification and extraction, whereas logical reasoning usually requires more steps. Summarization displays substantial variability across models—early for o3-mini but much later for Doubao-1.5-Thinking-Pro and DeepSeek-R1.

The long chain-of-thought reasoning paradigm provides a slight efficiency gain within families. For example, Claude-3.7-Sonnet-Thinking attains a lower average $P_{\text{offset}}$ than Claude-3.7-Sonnet, and Doubao-1.5-Thinking-Pro slightly improves over Doubao-1.5-Pro, while the DeepSeek pair is mixed. Across families, the reasoning-focused o3-mini also tends to learn earlier than GPT-4o. We find that $P_{\text{offset}}$ highlights systematic task effects and modest benefits from long-form reasoning, while revealing substantial cross-model variability in true test-time learning efficiency. Consistent with $P_{\text{first}}$, $P_{\text{offset}}$ varies widely across models and tasks and does not align with overall accuracy, underscoring that EvaLearn captures a dynamic learning dimension that is distinct from static capability.

Overall, all metrics comprehensively assess the learning performance from multiple perspectives. EvaLearn evaluates models in a more realistic and dynamic way, helping researchers better understand the gap between models and humans, and driving the development of more powerful models.

## 4   Conclusion

We present EvaLearn, a novel benchmark that sequentially evaluates the learning capability and efficiency of models within specific tasks. EvaLearn is equipped with a suite of comprehensive metrics, revealing significant performance differences among frontier models across diverse tasks, including both thinking-based and non-thinking-based models. Moreover, we find that while some models can effectively leverage teacher model feedback on previous solutions to enhance learning, others struggle to benefit from such feedback. EvaLearn offers a new perspective on assessing the potential of LLMs and serves as a pioneering step toward dynamic evaluation. The limitations and future work of EvaLearn are discussed in Appendix A, and related work is presented in Appendix B.

## Acknowledgments

We gratefully acknowledge the significant contributions made by the annotation teams at ByteDance, whose diligent work was essential to the success of this paper. The core members of the annotation team include Di Cheng, Linhua Deng, Yanxi Fu, Yafei Qiao, Chaoqian Ren, Mei Su, Ying Wu, Baitong Yang, and Xingyu Zhu. We wish to express our sincere appreciation to an undisclosed

third-party annotation company for their substantial support in data annotation. We also want to thank all individuals who participated in and supported this project for their valuable input. Finally, we wish to thank the AC and anonymous reviewers for their constructive comments. The authors wish to thank the anonymous reviewers for their helpful comments. This work was partially funded by the Science and Technology Commission of Shanghai Municipality (No. 24511103100), National Natural Science Foundation of China (No.62476061, 62441602, 62206057), Shanghai Rising-Star Program (23QA1400200), and Natural Science Foundation of Shanghai (23ZR1403500).

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

# A  Limitations and Future Work

EvaLearn is the first benchmark to quantitatively evaluate the rapid learning capabilities of LLMs on a single task. We discuss its limitations, the new insights it offers for the community, and the directions it opens for future research into the causes of learning differences between models and the development of learning methods.

**Towards including highly challenging open-ended problems.** EvaLearn focuses on problems that can be reliably evaluated with well-defined rubrics, ensuring trustworthy automated assessment. However, during the annotation process, we found that some highly challenging open-ended problems—where systematic instance-level rubrics are difficult to construct—are not yet included, even though such problems can be valuable for human learning and reasoning. In the future, we aim to develop new evaluation strategies that can effectively assess model responses to open-ended tasks, further expanding the task coverage of EvaLearn.

**Towards benchmarking more open-source LLMs.** We benchmark current frontier LLMs and reveal significant differences in their task learning performance. While these advanced models substantially outperform existing open-source models, there is still limited understanding of how open-source models compare in terms of task learning ability. In future work, we plan to extend our evaluation to a broader range of open-source models, aiming to better quantify this gap and foster progress within the open-source LLM community.

**Exploring the roots of learning capability and efficiency differences.** Our analysis quantifies the learning capability and efficiency of the models, and reveals that these are not strongly correlated with the models' static abilities. The fundamental factors driving differences in these aspects across models warrant deeper investigation. In future work, we plan to conduct more detailed studies to uncover the underlying causes of these disparities. We hope that EvaLearn will inspire the community to further explore and understand the mechanisms underlying LLM learning capability and efficiency.

**Broadening the exploration of learning methods.** EvaLearn has investigated the impact of two learning methods on both learning capability and efficiency. However, LLMs may benefit from a broader range of strategies, such as integration with retrieval-augmented generation (RAG) [23; 101; 60]. In future work, we will leverage EvaLearn to systematically analyze the effectiveness of additional learning methods in task learning scenarios. We also hope that EvaLearn will encourage the community to innovate and develop more effective approaches for rapid adaptation to new tasks.

# B  Related Work

**Evaluation Perspective and Paradigm.** Previous work on evaluating LLMs has mainly focused on evaluating their static capabilities [10; 39; 27; 95], such as factual knowledge [62; 29; 32; 97; 94; 80; 68], reasoning [30; 18; 100; 78; 37; 44; 103; 70; 50; 42], instruction following [96; 104; 61; 82; 34; 86; 16], and code generation [74; 13; 5]. Current evaluation paradigms can be broadly categorized into two types: parallel evaluation and interactive evaluation [22; 73; 69; 64; 17; 41; 91]. The vast majority of existing benchmarks adopt a parallel evaluation approach. This evaluation method only measures the static capabilities of models, providing little insight into their capability to learn or adapt dynamically over time [93; 71; 43; 7; 47; 40; 33; 85; 46; 99]. On the other hand, interactive evaluation is commonly used in multi-turn behavior assessments [102; 105; 12; 98; 31; 87; 35]. This paradigm emphasizes the dynamic evaluation of a model's ability to engage in multi-turn interactions, where tasks themselves remain i.i.d., but actions or states are temporally dependent. Interactive evaluation more closely reflects real-world scenarios [48; 84; 54; 52]. Our work also falls under the category of dynamic evaluation. However, unlike these methods, EvaLearn requires models to sequentially solve a series of problems within the same task, evaluating learning capability and efficiency, providing a direct indication of the model's learning potential. It fills a critical void in the current evaluation landscape.

**Learning Paradigms.** We compare two learning paradigms—learning from demonstrations [8; 53; 21] and learning from feedback [66]—in terms of their ability to stimulate model learning. Broadly, the former enables models to learn output formats and problem-solving strategies from demonstrations [49; 72; 24; 1; 19; 67; 83; 92; 14; 2], while the latter allows models to leverage experience gained from previous problems to improve subsequent performance [15; 88; 81; 89; 51; 11]. Both paradigms can be incorporated into model training to enhance overall capability [59; 63]. In this paper, we apply

both approaches at inference time, requiring models to sequentially solve a series of problems to assess whether they can benefit from prior canonical solutions or experience when tackling additional problems. Importantly, the evaluation metrics in EvaLearn are decoupled from the choice of learning method, ensuring the flexibility and general applicability.

## C  Broader Impact

In this work, we propose a pioneering benchmark, EvaLearn, which is designed to dynamically assess the learning ability and efficiency of LLMs. We hope EvaLearn provides a new evaluation perspective for assessing LLM potential and promotes the development of deeper and more dynamic evaluation approaches. All the data in EvaLearn is used to measure the learning capability of LLMs, which does not contain any biased information. We do not see any negative societal impacts of this work.

## D  License For Artifacts and Data Consent

All models used in this work, whether open-source or closed-source, are permitted for academic research. The licenses for DeepSeek-V3 and DeepSeek-R1 are MIT, while the license for Qwen2.5-32B-Instruct is Apache 2.0.

## E  Data Annotation Process

We adopt a hybrid approach combining human expertise and advanced LLMs for data annotation. The annotation process consists of three main stages.

**Problem Collection.** We begin with a collection of questions sourced from our model API platform, which are anonymized to remove any personally identifiable information or identification markers. Four annotators independently and carefully review each question, identifying those for which humans could potentially improve their performance through repeated practice on similar problems. Only questions unanimously judged as "learnable" by all annotators are retained as seed problems. These seed problems are then categorized into six task categories. Next, we use Claude-3.7 [4] to analyze each problem, identifying the required skill dimensions and suggesting potential directions for increasing problem complexity. With the aid of these model-generated strategies, nine annotators write additional similar and challenging problems for each seed question, expanding datasets with diverse variants.

**Problem filtering and canonical answer annotation.** To further increase the difficulty of the dataset, we leverage advanced LLMs to screen the seed problems. Specifically, for each question, we randomly select three LLMs from a model list (as shown in Table 3) to generate candidate answers. This approach helps avoid bias toward or against any specific model. Seven annotators then review these answers to write the correct canonical answer for each question, and assess the correctness of each model-generated response, providing explanations for their judgments. To ensure high data quality, an additional three annotators double-check both the written canonical answers and the explanations for the correctness judgments of the model responses. We retain only those challenging questions for which all three models fail to provide a correct answer, thereby maintaining a high level of difficulty among the seed problems.

All annotators involved in this work were fairly compensated in accordance with the labor standards of their respective countries.

**Instance-level rubric annotation and validation.** Eight annotators are responsible for writing rubrics for each retained challenging problem, which serve as criteria for evaluating the correctness of model responses. Annotators first determine whether each problem has a unique correct answer. For problems with a unique answer, the rubric is constructed as a concise summary of the canonical answer. For problems with multiple valid answers, annotators write rubrics that comprehensively cover all acceptable solutions, clearly specifying the required conditions for correctness. Overly open-ended problems that are difficult to evaluate (e.g., brainstorming tasks) are excluded from the dataset.

Each rubric is then validated by two annotators. Specifically, for every problem, three models are randomly selected from the model list to generate three candidate answers. Annotators assess whether

the rubric can accurately evaluate all responses and provide explanations. Rubrics that fail this validation step are revised and re-evaluated. An additional three annotators conduct a final quality check on all fields, including the canonical answers, the explanations for the correctness judgments of the model responses, and the rubrics. Finally, we construct problem sequences from the retained set of challenging problems. Each sequence consists of problems from the same task category, and the order of problems within each sequence is randomized.

Table 3: The model list used for ensuring the difficulty of datasets and validating the effectiveness of instance-level rubrics.

| Model | Published date |
|---|---|
| Claude-3.7-Sonnet [4] | February 25, 2025 |
| Claude-3.7-Sonnet-Thinking [4] | February 25, 2025 |
| Claude-3.5-Sonnet [3] | June 21, 2024 |
| DeepSeek-V3 [45] | December 26, 2024 |
| DeepSeek-R1 [26] | January 15, 2025 |
| Doubao-1.5-Pro-32k [9] | May 9, 2025 |
| Doubao-1.5-Thinking-Pro [65] | April 17, 2025 |
| Doubao-1.5-Pro-256k [9] | May 9, 2025 |
| Qwen2.5-7b-Instruct [90] | September 19, 2024 |
| Qwen2.5-32b-Instruct [90] | September 19, 2024 |
| QwQ-32b [75] | March 6, 2025 |
| OpenAI-o3-mini [57] | January 31, 2025 |
| OpenAI-o1 [56] | September 12, 2024 |
| OpenAI-o4-mini [58] | April 16, 2025 |

## F   List of Evaluation Models

In this work, we comprehensively evaluate the learning capability and efficiency of nine state-of-the-art LLMs. These LLMs contain:

**(1) Claude-3.7-Sonnet** [4] is the most intelligent model released by Anthropic to date, particularly in terms of coding capabilities, which is an upgraded version of Claude 3.5 Sonnet.

**(2) Claude-3.7-Sonnet-Thinking** [4] is the extended thinking mode of Claude-3.7-Sonnet, which performs self-reflection before answering in order to improve performance on tasks such as math, physics, instruction-following, coding, and many others.

**(3) DeepSeek-V3** [45] is a strong Mixture-of-Experts (MoE) language model with 671B total parameters with 37B activated for each token.

**(4) DeepSeek-R1** [26] is the first-generation reasoning model developed by DeepSeek AI, which incorporates multi-stage training and cold-start data before reinforcement learning, enabling it to demonstrate strong reasoning capabilities.

**(5) Doubao-1.5-Pro** [9] is a high-performance sparse MoE large language model that focuses on achieving an optimal balance between inference performance and model capability developed by the Doubao team.

**(6) Doubao-1.5-Thinking-Pro** [65] shares the same model base as Seed-Thinking-v1.5, which is an advancing superb reasoning model with reinforcement learning. It is a MoE model with a relatively small size, featuring 20B activated and 200B total parameters. It is capable of reasoning through thinking before responding, resulting in improved performance on a wide range of benchmarks.

**(7) Qwen2.5-32b-Instruct** [90] is the instruction-tuned 32B version of the Qwen2.5 series, which is an improved iteration based on the Qwen2 family.

**(8) GPT-4o** [55] is a state-of-the-art LLM developed by OpenAI, which demonstrates strong performance across a wide range of tasks, including natural language understanding, complex reasoning, instruction following, code generation, and problem solving. It achieves competitive results on

Table 4: Results of $P_{\text{offset}}$. Full task names for the abbreviations can be found in Section 2.1. Blue-colored cells indicate the best performance for each task category.

| Model | Sum | Cla | Ex | LR | MR | SR | Overall |
|---|---|---|---|---|---|---|---|
| **Non-thinking-based** | | | | | | | |
| DeepSeek-V3 | 3.82 | 4.0 | 3.88 | 5.27 | 3.27 | 4.00 | 4.04 |
| Claude-3.7-Sonnet | 4.08 | 3.08 | 3.71 | 5.69 | 3.24 | 1.94 | 3.62 |
| GPT-4o | 4.08 | 3.45 | 3.24 | 6.0 | 2.41 | 3.29 | 3.75 |
| Doubao-1.5-Pro | 3.54 | 3.58 | 3.71 | 6.75 | 3.24 | 3.06 | 3.98 |
| Qwen2.5-32b-Instruct | 3.93 | 3.69 | 3.35 | 6.38 | 3.65 | 3.53 | 4.09 |
| **Thinking-based** | | | | | | | |
| OpenAI-o3-mini | 2.6 | 3.42 | 3.71 | 2.93 | 3.4 | 3.62 | 3.28 |
| Doubao-1.5-Thinking-Pro | 4.9 | 3.83 | 2.88 | 3.44 | 3.67 | 4.81 | 3.92 |
| DeepSeek-R1 | 4.7 | 4.5 | 2.76 | 4.7 | 3.62 | 4.19 | 4.08 |
| Claude-3.7-Sonnet-Thinking | 3.8 | 2.77 | 3.18 | 5.55 | 3.41 | 2.53 | 3.54 |
| Gemini-2.5-Pro | 4.31 | 3.0 | 3.47 | 3.66 | 2.71 | 3.46 | 3.44 |
| Gemini-2.5-Flash | 4.08 | 3.0 | 2.76 | 6.55 | 3.4 | 3.18 | 3.83 |

challenging benchmarks in mathematics, logic, and programming, and maintains high performance across diverse evaluation settings.

**(9) OpenAI-o3-mini** [57] is a cost-efficient model in the reasoning series, previewed in December 2024, which pushes the boundaries of what small models can achieve. It delivers exceptional STEM capabilities—with particular strengths in science, math, and coding—while maintaining the low cost and reduced latency of OpenAI-o1-mini.

All models are evaluated via their respective APIs. For non-thinking-based models, the temperature is set to 0.2, while thinking-based models do not support temperature adjustment. Moreover, both input and output lengths are set to the maximum supported by each model.

The total cost for running all experiments in our study was approximately $4,357.14, which includes the evaluation of all models across all tasks and learning paradigms on the entire benchmark.

# G    Additional Experimental Results

## G.1    Case Study: Do LLMs truly utilize prior judgments to solve problems?

### Case Study 1: Comparison with Parallel Solving (Zero-shot) and Sequence Solving (Feedback Learning)

Figure 4 illustrates how model performance differs between the parallel solving (zero-shot) and sequence solving (feedback learning) paradigms in a sequential reasoning task involving maze navigation. In the zero-shot setting (Response A), Claude-3.7-Sonnet makes a critical directional error in the final segment, incorrectly assuming the direction after a turn and arriving at the wrong coordinate $(7, 4)$. This highlights the challenge of maintaining spatial orientation without feedback.

In contrast, under the feedback learning paradigm (Response B), the same model correctly tracks all movements and reaches the accurate exit coordinate $(7, 12)$. The model explicitly defines directional conventions at the outset ("Let me define directions first...") and methodically tracks orientation after each turn, demonstrating clear evidence of learning from previous feedback on similar problems.

### Case Study 2: Evidence of learning from prior experience

Figure 5 demonstrates how models utilize prior experience in sequential reasoning tasks. In this case, DeepSeek-R1's thinking process while solving a 4x4 sliding puzzle explicitly references knowledge gained from previous problems.

Table 5: Comparison of overall accuracy across four solving methods, including two parallel methods (i.e., zero-shot and few-shot) and two sequential methods (i.e., demonstration learning and feedback learning). **Sum** denotes the summarization task. **Cla** denotes the classification task. **Ex** denotes the extraction task. **LR** denotes the logical reasoning task. **MR** denotes the mathematical reasoning task. **SR** denotes the sequential reasoning task. All values are shown as %.

| Model | Paradigm | Cla | Ex | LR | MR | SR | Sum | Overall |
|---|---|---|---|---|---|---|---|---|
| *Non-thinking-based* | | | | | | | | |
| Doubao-1.5-Pro | Zero-shot | 70.8 | **45.0** | 10.3 | 61.7 | **41.7** | 71.7 | **31.3** |
| | Few-shot | 60.0 | 35.1 | 8.4 | 54.4 | 31.6 | 63.2 | 25.7 |
| | Demonstration Learning | **71.4** | 43.7 | 8.8 | 63.9 | 28.6 | 71.4 | 29.0 |
| | Feedback Learning | 68.1 | 42.9 | **10.8** | 71.4 | 36.1 | 71.4 | 31.2 |
| DeepSeek-V3 | Zero-shot | 72.9 | **45.0** | 25.6 | 71.7 | **60.0** | 81.7 | **43.5** |
| | Few-shot | **75.6** | 40.4 | 23.0 | 61.4 | 57.9 | 78.9 | 40.0 |
| | Demonstration Learning | 70.3 | 40.3 | 17.6 | 68.9 | 51.3 | 74.1 | 36.4 |
| | Feedback Learning | 74.7 | 36.1 | 24.9 | 74.8 | 54.6 | 76.8 | 41.5 |
| Claude-3.7-Sonnet | Zero-shot | 64.6 | **48.3** | 8.1 | 48.3 | 33.3 | 76.7 | 28.4 |
| | Few-shot | **80.0** | 45.6 | 10.4 | 56.1 | 43.9 | 70.2 | 31.1 |
| | Demonstration Learning | 64.8 | 42.9 | 10.8 | 61.3 | 43.7 | 75.0 | 31.1 |
| | Feedback Learning | 75.8 | 46.2 | **15.5** | 63.9 | 49.6 | 75.0 | **35.6** |
| Qwen2.5-32b-Instruct | Zero-shot | 60.4 | 31.7 | 6.9 | 46.7 | 21.7 | 66.7 | 23.8 |
| | Few-shot | 60.0 | 31.6 | 6.4 | 38.6 | 24.6 | 66.7 | 22.5 |
| | Demonstration Learning | **69.2** | **34.5** | 7.4 | 43.7 | 21.8 | 64.3 | 24.1 |
| | Feedback Learning | 62.6 | 30.3 | **9.8** | 49.6 | 30.3 | 59.8 | **25.5** |
| GPT-4o | Zero-shot | 70.8 | **45.0** | 15.6 | 43.3 | 31.7 | 81.7 | 32.6 |
| | Few-shot | **77.8** | 40.4 | 15.4 | 54.4 | **40.4** | 70.2 | **32.9** |
| | Demonstration Learning | 71.4 | 35.3 | 8.5 | 56.3 | 34.5 | 75.0 | 28.3 |
| | Feedback Learning | 73.6 | 48.7 | 12.9 | 61.3 | 32.2 | 78.6 | 32.7 |
| *Thinking-based* | | | | | | | | |
| DeepSeek-R1 | Zero-shot | **89.6** | 48.3 | **41.7** | 78.3 | **66.7** | 86.7 | **55.7** |
| | Few-shot | 75.6 | **49.1** | 23.5 | 73.7 | 54.4 | 78.9 | 41.9 |
| | Demonstration Learning | 82.4 | 48.7 | 31.4 | **79.8** | 52.9 | 88.4 | 48.2 |
| | Feedback Learning | 79.1 | 48.7 | 29.6 | 74.8 | 51.3 | **89.3** | 46.4 |
| Claude-3.7-Sonnet-Thinking | Zero-shot | 66.7 | 43.3 | 12.5 | 46.7 | 31.7 | **86.7** | 31.2 |
| | Few-shot | 60.0 | 42.1 | 13.4 | 43.9 | **43.9** | 77.2 | 30.6 |
| | Demonstration Learning | 74.7 | **48.7** | 16.1 | **59.7** | 42.0 | 77.7 | 35.2 |
| | Feedback Learning | **80.2** | 48.7 | **18.8** | 58.0 | 46.2 | 78.6 | **37.4** |
| OpenAI-o3-mini | Zero-shot | 64.6 | 48.3 | 45.8 | 73.3 | 73.3 | 65.0 | 54.3 |
| | Few-shot | **80.0** | 50.9 | 50.7 | 75.4 | **84.2** | 73.7 | 60.0 |
| | Demonstration Learning | 73.6 | **51.3** | 53.1 | 78.2 | 73.9 | 67.9 | 60.0 |
| | Feedback Learning | 73.6 | 47.1 | **59.9** | 80.7 | 78.2 | **75.9** | **64.8** |
| Doubao-1.5-Thinking-Pro | Zero-shot | 79.2 | **55.0** | 42.5 | 85.0 | 73.3 | 85.0 | **57.1** |
| | Few-shot | **82.2** | 49.1 | 38.9 | 71.9 | 64.9 | 73.7 | 51.4 |
| | Demonstration Learning | 73.6 | 52.1 | 42.0 | 77.3 | 64.7 | 83.9 | 54.3 |
| | Feedback Learning | 70.3 | 52.9 | 39.4 | 77.3 | 55.5 | 82.1 | 51.6 |

The highlighted portions of the thinking process show: "After struggling, I think the correct sequence should be similar to the user's previous solution, involving multiple 'Up' moves followed by 'Right' and 'Down'." The model further reasons: "The correct answer might involve a series of 'Up' moves to position 'M' at the top, then moving other numbers into place."

Ultimately, the model applies this learned pattern to formulate an 11-step solution: ["Up", "Up", "Up", "Right", "Down", "Down", "Down", "Right", "Up", "Up", "Up"], well within the 12-move limit.

This case illustrates how models can extract patterns from prior experience and apply them to new related problems, demonstrating genuine experiential learning capability.

---

**The case used for Case Study 1**

**Task Type:** Sequential Reasoning
**Model:** Claude-3-7-sonnet-20250219
**Question:** You enter a $12 \times 12$ maze. The coordinates on this maze are represented by $(x, y)$, where $x$ is horizontal (left and right directions) and $y$ is vertical (up and down directions). The bottom left square is at coordinates $(1, 1)$ and the top right square is at coordinates $(12, 12)$. You enter the maze at coordinates $(3, 1)$. Following are the movements you take to reach the exit, where you will give the coordinates after each movement, as well as the coordinates of the final exit point, which can be anywhere on any coordinate of the maze:
After entry, you take four steps forward, turn right and take two steps forward and another right and 3 steps forward, where you hit a wall. After turning back around and taking a step, you turn right again and take 3 steps forward, turn left and take 5 steps forward, left again and 2 steps forward, where you hit a wall again. You take a step back, and turn right, and take 4 steps forward to reach the exit.
**Rubric:** Student answers must meet requirements including but not limited to the following:
1. The answer must clearly state that the maze exit is at $(7, 12)$, otherwise it receives a score of 0;
2. Correctly understand the coordinate system: $x$ represents the horizontal direction, $y$ represents the vertical direction, with the bottom left corner at $(1, 1)$ and the top right corner at $(12, 12)$;
3. Accurately identify the starting coordinates $(3, 1)$;
4. Correctly track all movements and calculate coordinates after each movement:
- Coordinates after four steps forward from the entrance $(3, 5)$;
- Coordinates after turning right and taking two steps $(5, 5)$;
- Coordinates after turning right again and taking three steps (hitting a wall) $(5, 2)$;
- Coordinates after turning around and taking one step back $(5, 3)$;
- Coordinates after turning right and taking three steps $(8, 3)$;
- Coordinates after turning left and taking five steps $(8, 8)$;
- Coordinates after turning left again and taking two steps (hitting a wall again) $(6, 8)$;
- Coordinates after taking one step back $(7, 8)$;
- Final coordinates (exit) after turning right and taking four steps $(7, 12)$;
5. Correctly interpret directional terms: forward, right turn, left turn, turn around, step back, etc.;
6. Accurately handle wall situations: stop moving when hitting a wall, then change direction as instructed; do not turn around without specific instructions to do so.
**Response A [Parallel Soving, Zero-shot] (Incorrect):**
Let me track your movements through the maze, step by step, starting at coordinates $(3, 1)$.
Entry point: $(3, 1)$
1. "Four steps forward" - I'll assume "forward" means moving upward (increasing $y$-coordinate) $(3, 1) \rightarrow (3, 2) \rightarrow (3, 3) \rightarrow (3, 4) \rightarrow (3, 5)$Current position: $(3, 5)$
2. "Turn right and take two steps forward" Right turn means moving east (increasing $x$-coordinate) $(3, 5) \rightarrow (4, 5) \rightarrow (5, 5)$ Current position: $(5, 5)$
3. "Another right and 3 steps forward" Right turn means moving south (decreasing $y$-coordinate) $(5, 5) \rightarrow (5, 4) \rightarrow (5, 3) \rightarrow (5, 2)$ Current position: $(5, 2)$
4. "Hit a wall, turn back around and take a step" Turn around means moving north (increasing $y$-coordinate) $(5, 2) \rightarrow (5, 3)$ Current position: $(5, 3)$

5. "Turn right again and take 3 steps forward" Right turn means moving east (increasing $x$-coordinate) $(5,3) \rightarrow (6,3) \rightarrow (7,3) \rightarrow (8,3)$ Current position: $(8,3)$
6. "Turn left and take 5 steps forward" Left turn means moving north (increasing $y$-coordinate) $(8,3) \rightarrow (8,4) \rightarrow (8,5) \rightarrow (8,6) \rightarrow (8,7) \rightarrow (8,8)$ Current position: $(8,8)$
7. "Left again and 2 steps forward" Left turn means moving west (decreasing $x$-coordinate) $(8,8) \rightarrow (7,8) \rightarrow (6,8)$ Current position: $(6,8)$
8. "Hit a wall, take a step back, and turn right" Step back means moving east (increasing $x$-coordinate) $(6,8) \rightarrow (7,8)$ Current position: $(7,8)$ Right turn means facing south
9. "Take 4 steps forward to reach the exit" Moving south (decreasing $y$-coordinate) $(7,8) \rightarrow (7,7) \rightarrow (7,6) \rightarrow (7,5) \rightarrow (7,4)$
Therefore, the exit coordinates are $(7,4)$.

**Response B [Sequential Solving, Feedback Learning, Position in Sequence: 3](Correct):**
To solve this maze navigation problem, I need to carefully track the coordinates after each movement, starting from the entry point at $(3,1)$.
Let me define directions first:
- Initially, I'll assume we're facing north (up on the $y$-axis)
- Forward means moving in the direction we're facing
- Right means turning $90°$ clockwise
- Left means turning $90°$ counterclockwise
Starting at $(3,1)$ facing north:
1. Four steps forward: $(3,1) \rightarrow (3,2) \rightarrow (3,3) \rightarrow (3,4) \rightarrow (3,5)$ - Now at $(3,5)$
2. Turn right (now facing east) - Still at $(3,5)$
3. Two steps forward: $(3,5) \rightarrow (4,5) \rightarrow (5,5)$ - Now at $(5,5)$
4. Turn right again (now facing south) - Still at $(5,5)$
5. Three steps forward: $(5,5) \rightarrow (5,4) \rightarrow (5,3) \rightarrow (5,2)$ - Now at $(5,2)$
6. Hit a wall, turn back around (now facing north) - Still at $(5,2)$
7. Take a step: $(5,2) \rightarrow (5,3)$ - Now at $(5,3)$
8. Turn right (now facing east) - Still at $(5,3)$
9. Three steps forward: $(5,3) \rightarrow (6,3) \rightarrow (7,3) \rightarrow (8,3)$ - Now at $(8,3)$
10. Turn left (now facing north) - Still at $(8,3)$
11. Five steps forward: $(8,3) \rightarrow (8,4) \rightarrow (8,5) \rightarrow (8,6) \rightarrow (8,7) \rightarrow (8,8)$ - Now at $(8,8)$
12. Turn left again (now facing west) - Still at $(8,8)$
13. Two steps forward: $(8,8) \rightarrow (7,8) \rightarrow (6,8)$ - Now at $(6,8)$
14. Hit a wall, take a step back: $(6,8) \rightarrow (7,8)$ - Now at $(7,8)$
15. Turn right (now facing north) - Still at $(7,8)$
16. Four steps forward: $(7,8) \rightarrow (7,9) \rightarrow (7,10) \rightarrow (7,11) \rightarrow (7,12)$ - Now at $(7,12)$
The final exit point coordinates are $(7,12)$.

Figure 4: The case used in Case Study 1 is designed for comparison between Parallel Solving (Zero-shot) and Sequence Solving (Feedback Learning).

**Task Type:** Logical Reasoning
**Model:** Deepseek-R1
**Question:** You are a virtual expert in solving a 4x4 sliding puzzle problem. Please follow the instructions and rules below to complete the solving. Your goal is to reach the goal state with valid moves.
[The goal state]

| % | 1 | j | i |
|---|---|---|---|
| = | 8 | ? | f |
| * | F | n | Z |
| N | M | 0 | t |

[Instructions]
The puzzle consists of a 4x4 grid containing various symbols, with 'M' representing the movable space. Only 'M' can be moved horizontally or vertically, and the objective is to reach the goal state from a given initial state.
[Rules]
1. Only 'M' can be moved horizontally or vertically.
2. Each move is chosen from the following set of options:
- 'Left': move 'M' to the left
- 'Down': move 'M' downward
- 'Right': move 'M' to the right
- 'Up': move 'M' upward
3. The next move must be chosen from the valid move set depending on the position of 'M'.
For example:

| p1 | p2 | p3 | p4 |
|---|---|---|---|
| p5 | p6 | p7 | p8 |
| p9 | p10 | p11 | p12 |
| p13 | p14 | p15 | p16 |

- If 'M' is located at position 'p1', the valid move set is ['Down', 'Right'].
- If 'M' is located at position 'p2', the valid move set is ['Down', 'Left', 'Right'].
- If 'M' is located at position 'p3', the valid move set is ['Down', 'Left', 'Right'].
- If 'M' is located at position 'p4', the valid move set is ['Down', 'Left'].
- If 'M' is located at position 'p5', the valid move set is ['Up', 'Down', 'Right'].
- If 'M' is located at position 'p6', the valid move set is ['Up', 'Down', 'Left', 'Right'].
- If 'M' is located at position 'p7', the valid move set is ['Up', 'Down', 'Left', 'Right'].
- If 'M' is located at position 'p8', the valid move set is ['Up', 'Down', 'Left'].
- If 'M' is located at position 'p9', the valid move set is ['Up', 'Down', 'Right'].
- If 'M' is located at position 'p10', the valid move set is ['Up', 'Down', 'Left', 'Right'].
- If 'M' is located at position 'p11', the valid move set is ['Up', 'Down', 'Left', 'Right'].
- If 'M' is located at position 'p12', the valid move set is ['Up', 'Down', 'Left'].
- If 'M' is located at position 'p13', the valid move set is ['Up', 'Right'].
- If 'M' is located at position 'p14', the valid move set is ['Up', 'Left', 'Right'].
- If 'M' is located at position 'p15', the valid move set is ['Up', 'Left', 'Right'].
- If 'M' is located at position 'p16', the valid move set is ['Up', 'Left'].
4. Diagonal moves are not allowed.
5. The objective is to return the moves which can reach the goal state.
Please solve the following problem. Your answer should be in the format of a list of moves, for example ['Left', 'Up', 'Up'].
This problem can be solved within 1 to 12 steps. The next move must be chosen from the valid move set ['Left', 'Right', 'Up', 'Down']. The maximum step number you can take is 12. Try to reach the goal state using the least number of steps (<= 12). **DO NOT exceed 12 steps.**

[Initial State]:
```
=   %   j   i
*   1   ?   f
N   M   n   Z
F   8   0   t
```
**Thinking Process:**

Okay, let's tackle this new sliding puzzle problem. The goal state is given, and the movable tile is 'M'. The initial state is:
```
=   %   j   i
*   1   ?   f
N   M   n   Z
F   8   0   t
```

First, I need to compare the initial state with the goal state to identify the differences. The goal state is:
```
%   1   j   i
=   8   ?   f
*   F   n   Z
N   M   0   t
```

Looking at the positions, let's break it down row by row.
...
This seems complex. Let's try to approach step by step.
...
After struggling, I think the correct sequence involves moving 'M' up to facilitate the movement of '8' and '1'. The correct answer provided by the user for the previous problem involved multiple 'Up' moves followed by 'Right' and then 'Down's. Maybe a similar pattern applies here.
Perhaps the correct sequence is: ["Up", "Up", "Up", "Right", "Down", "Down", "Down", "Right", "Up", "Up", "Up"]
But adjusted for this problem. However, I need to verify. ...
This approach is not working. Given the time I've spent and the complexity, I think the correct answer might involve a series of 'Up' moves to position 'M' in the top rows, then moving '8' and '1' into place. Considering the user's previous correct answer had multiple 'Up's followed by 'Right' and 'Down's, maybe a similar pattern applies here.
After careful consideration, the correct sequence is likely: ["Up", "Right", "Up", "Right", "Up", "Left", "Down", "Left", "Up", "Right", "Down", "Left"]
But adjusted for this problem. However, without more time, I'll settle on the following steps based on moving 'M' up to facilitate tile movements: ...
But this requires more steps. Given the maximum of 12 steps, the answer is likely: ["Up", "Up", "Up", "Right", "Down", "Down", "Down", "Right", "Up", "Up", "Up"]

**Response:**

To solve the puzzle, I need to move the 'M' tile strategically to rearrange the other tiles into their goal positions.
First, I'll analyze the initial and goal states:
Initial state:
```
=   %   j   i
*   1   ?   f
N   M   n   Z
F   8   0   t
```
Goal states
```
%   1   j   i
=   8   ?   f
*   F   n   Z
N   M   0   t
```

Starting with 'M' at position p10, I'll execute the following sequence of moves:
1. Move Up: 'M' moves from p10 to p6, swapping with '1' - New state: = % j i / * M ? f / N 1 n Z / F 8 0 t

2. Move Up: 'M' moves from p6 to p2, swapping with '%' - New state: = M j i / * % ? f / N 1 n Z / F 8 0 t
3. Move Up: Since p2 is in the top row, moving up would be invalid. However, I can move left to p1, swapping with '=' - New state: M % j i / * = ? f / N 1 n Z / F 8 0 t
4. Move Right: 'M' moves from p1 to p2, swapping with '%' - New state: % M j i / * = ? f / N 1 n Z / F 8 0 t
5. Move Down: 'M' moves from p2 to p6, swapping with '=' - New state: % 1 j i / * M ? f / N = n Z / F 8 0 t
6. Move Down: 'M' moves from p6 to p10, swapping with '=' - New state: % 1 j i / * = ? f / N M n Z / F 8 0 t
7. Move Down: 'M' moves from p10 to p14, swapping with '8' - New state: % 1 j i / * = ? f / N 8 n Z / F M 0 t
8. Move Right: 'M' moves from p14 to p15, swapping with '0' - New state: % 1 j i / * = ? f / N 8 n Z / F 0 M t
9. Move Up: 'M' moves from p15 to p11, swapping with 'n' - New state: % 1 j i / * = ? f / N 8 M Z / F 0 n t
10. Move Up: 'M' moves from p11 to p7, swapping with '?' - New state: % 1 j i / * = M f / N 8 ? Z / F 0 n t
11. Move Up: 'M' moves from p7 to p3, swapping with 'j' - New state: % 1 M i / * = ? f / N 8 j Z / F 0 n t
The optimal solution is: ["Up", "Up", "Up", "Right", "Down", "Down", "Down", "Right", "Up", "Up", "Up"]
This sequence of 11 moves successfully transforms the initial state into the goal state, staying within the 12-move limit.

Figure 5: The case used in Case Study 2 provides evidence of learning from prior experience.

## G.2  Results of Overall sequence accuracy (Acc)

Figure 6 presents the overall accuracy results for four problem-solving methods: two parallel methods (i.e., zero-shot and few-shot) and two sequential methods (i.e., demonstration learning and feedback learning).

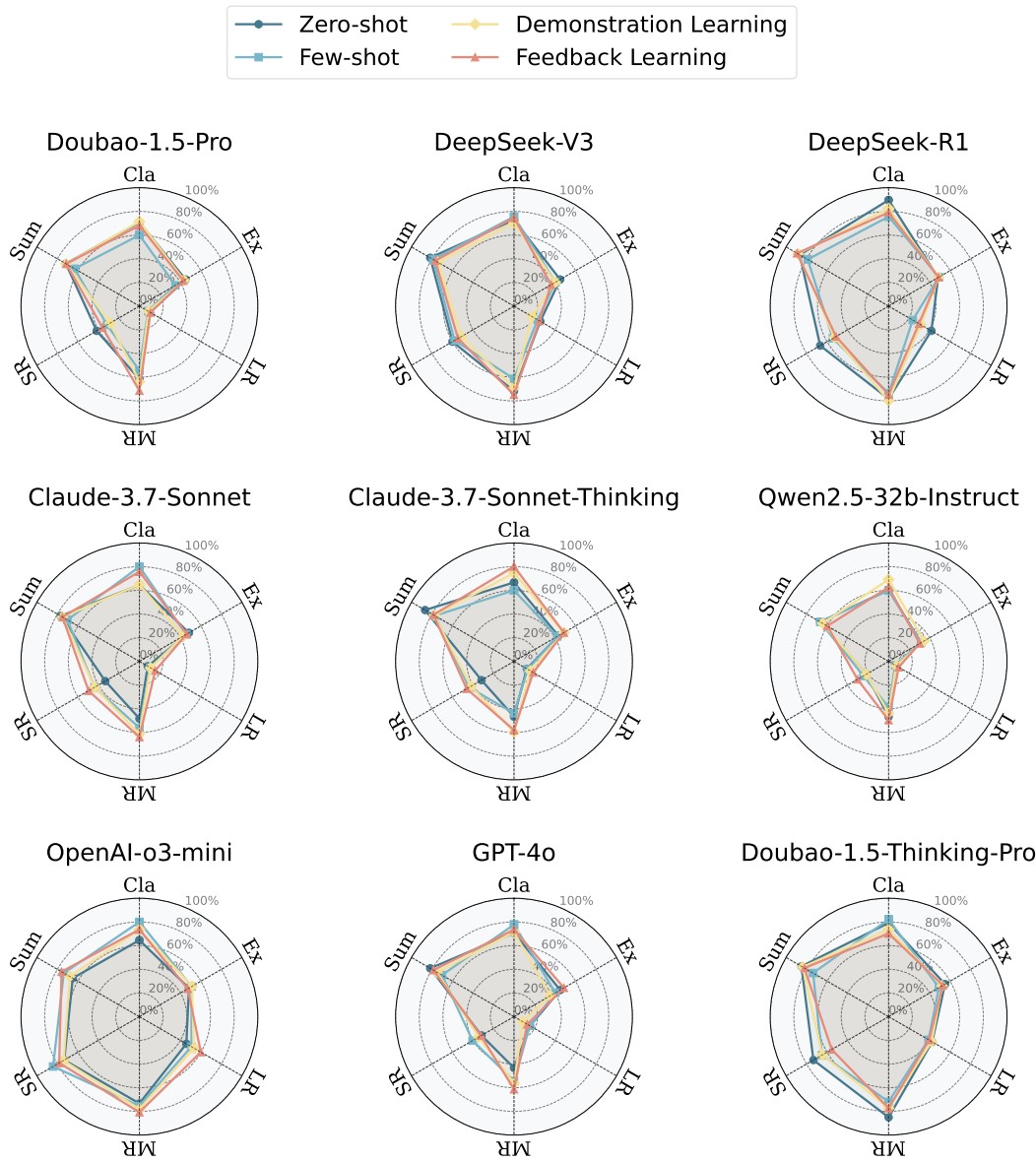

Figure 6: Comparison of overall accuracy across four solving methods, including two parallel methods (i.e., zero-shot and few-shot) and two sequential methods (i.e., demonstration learning and feedback learning). Full task names for the abbreviations can be found in Section 2.1.

### G.3 Results of Slope of fitted accuracy curve $k$

Figures 7 and 8 present the slope ($k$) of the fitted position-wise accuracy curve for the feedback learning and demonstration learning methods, respectively. Figure 9 further shows the difference in slopes ($\Delta_k$) between feedback learning and demonstration learning at corresponding positions. This metric quantifies the model's learning speed across all sequences by fitting a straight line to the position-wise accuracy curve using least squares regression.

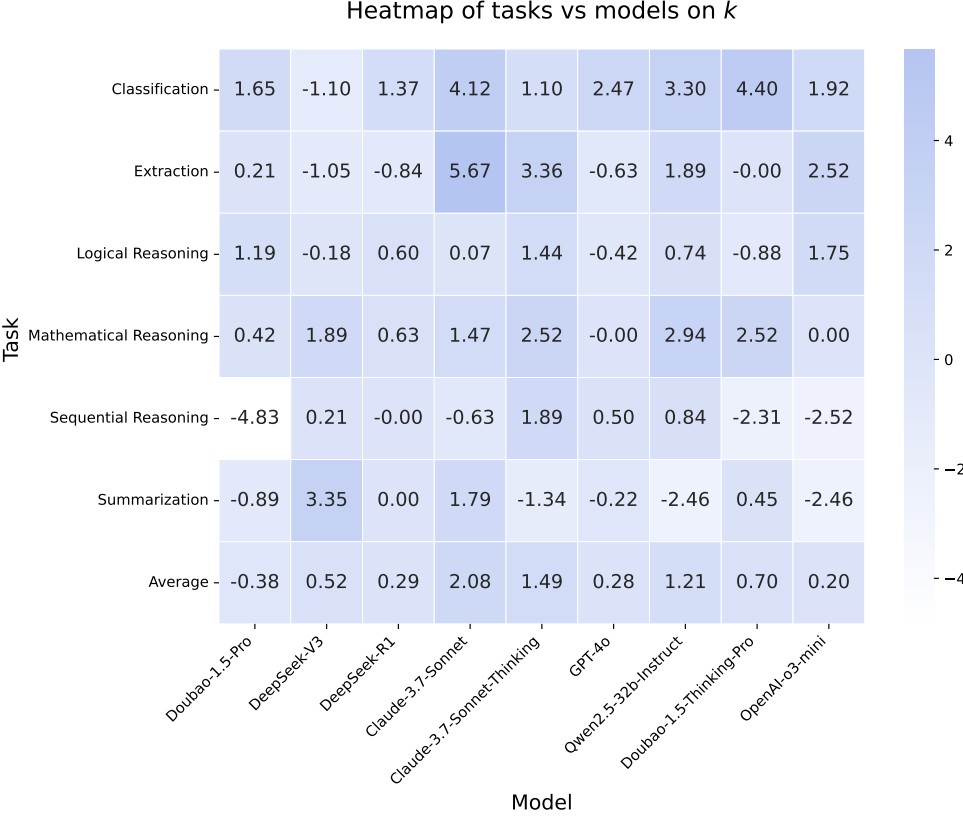

Figure 7: Results of the fitted position-wise accuracy curve slope ($k$) for feedback learning, across all models and tasks.

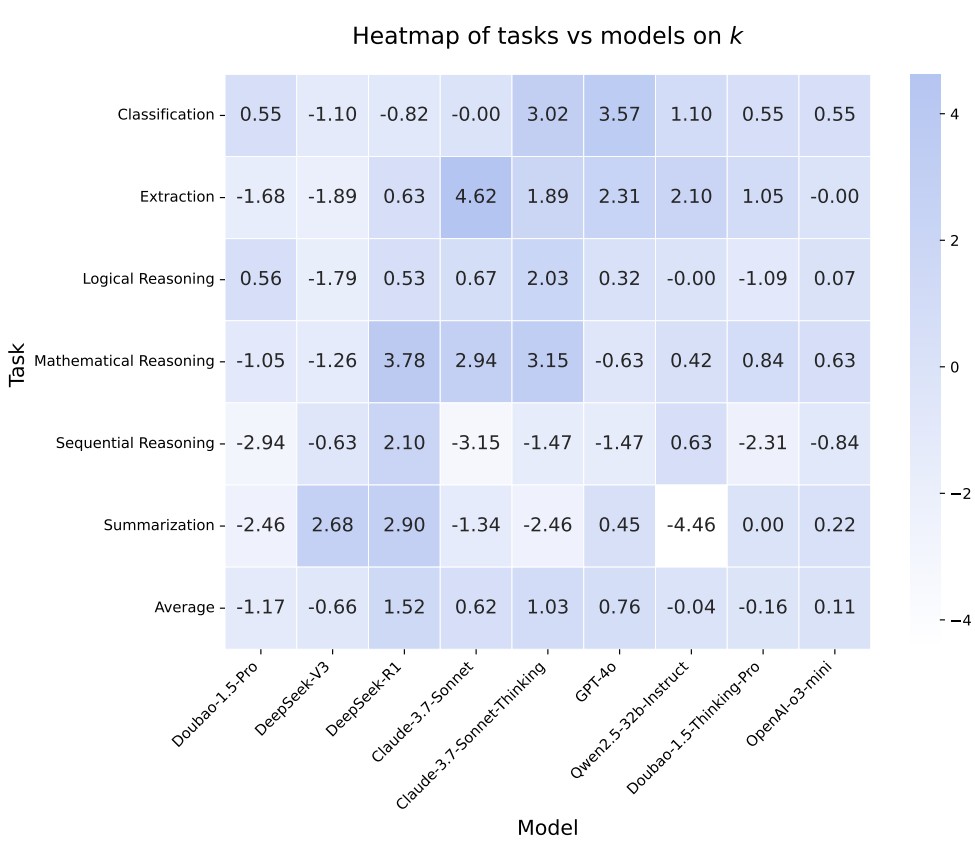

Figure 8: Results of the fitted position-wise accuracy curve slope ($k$) for demonstration learning, across all models and tasks.

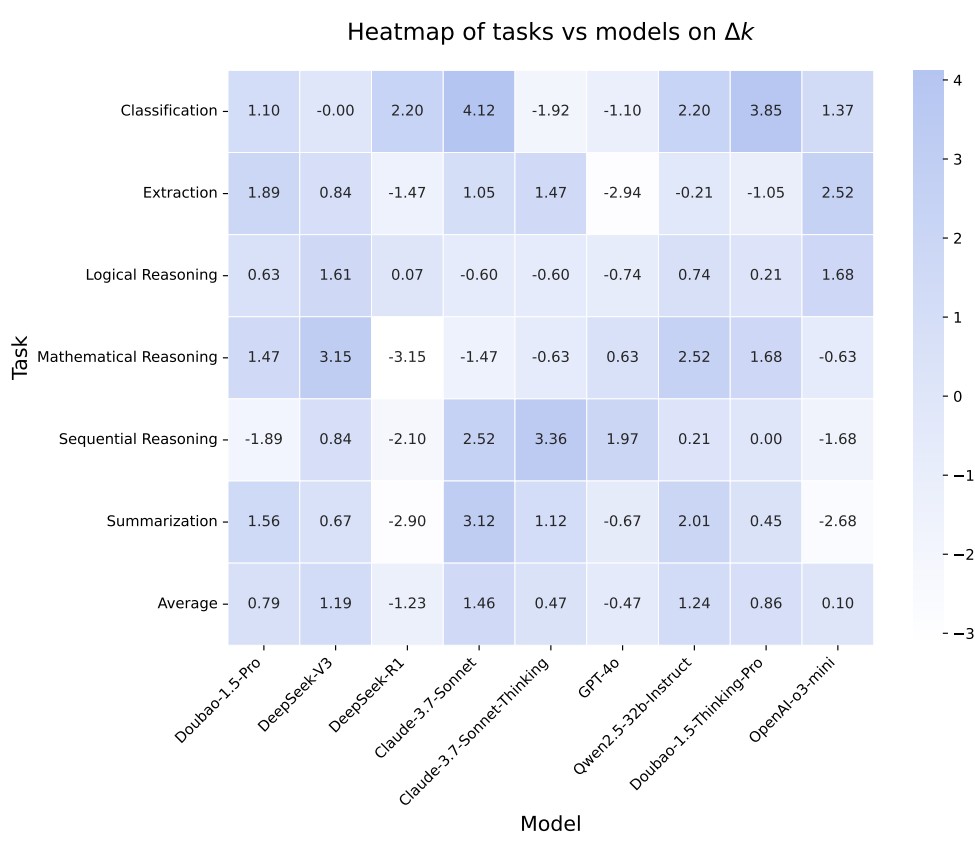

Figure 9: The difference in slopes ($\Delta_k$) between feedback learning and demonstration learning at corresponding positions, across all models and tasks.

## G.4 Results of Average Position of First Correct Solution ($P_{\text{first}}$)

Figures 10 and 11 show the results of the average position of the first correct solution ($P_{\text{first}}$) for feedback learning and demonstration learning, respectively. This metric measures how quickly, on average, the model achieves its first success in a sequence, thus indicating its initial learning speed.

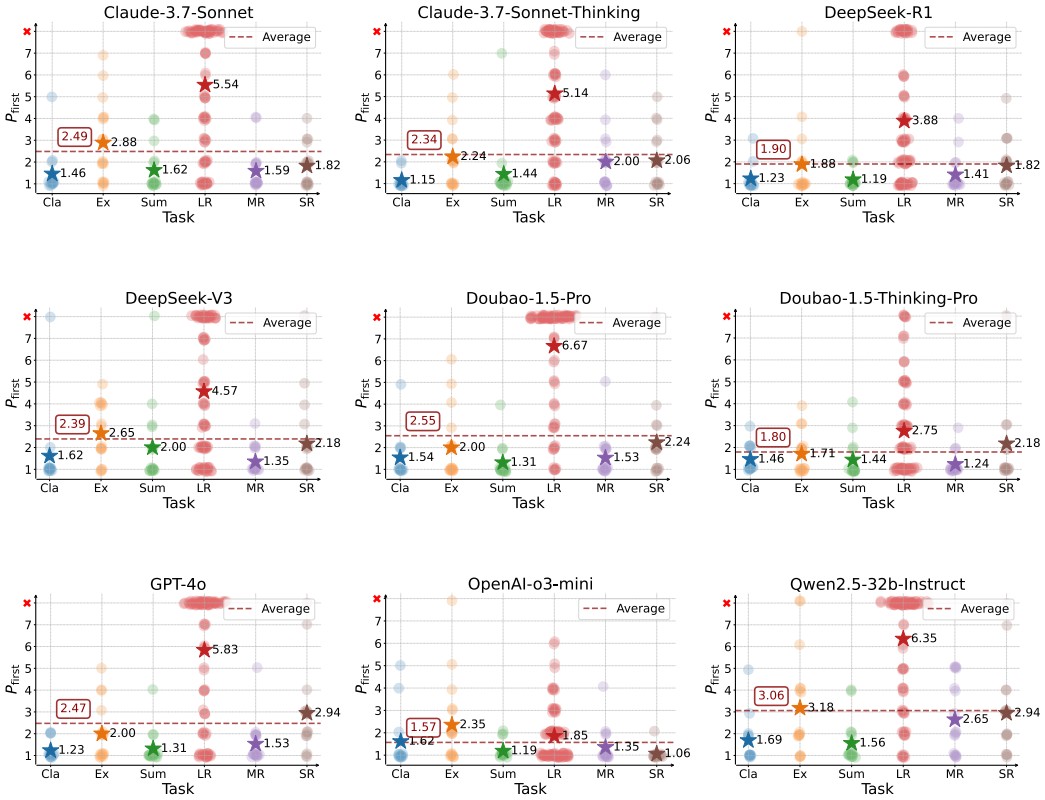

Figure 10: Results of the average position of the first correct solution $P_{\text{first}}$ for feedback learning.

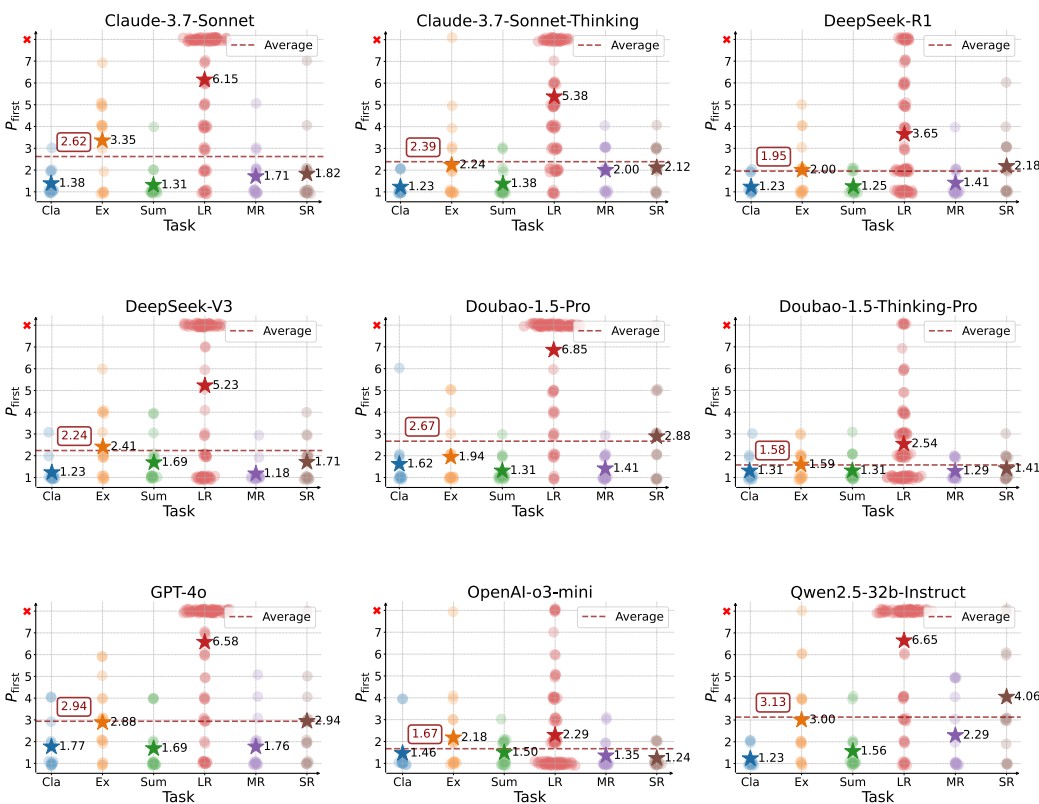

Figure 11: Results of the average position of the first correct solution $P_{\text{first}}$ for demonstration learning.

## G.5 Results of Average Number of Consecutive Correct Solutions ($N_{\text{consec}}$)

Figures 12 and 13 report the results of the average number of consecutive correct solutions ($N_{\text{consec}}$) for feedback learning and demonstration learning, respectively. This metric reflects the model's ability to leverage experience to consistently solve problems within a sequence, resulting in fewer errors.

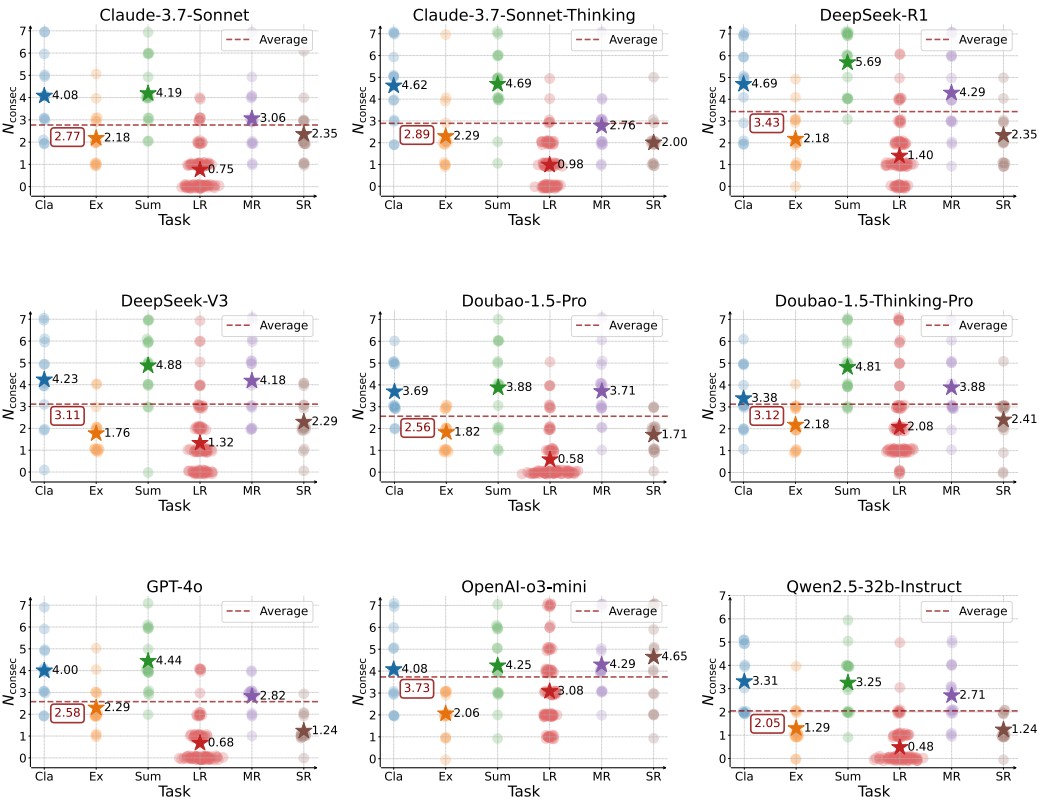

Figure 12: Results of the average number of consecutive correct solutions ($N_{\text{consec}}$) for feedback learning.

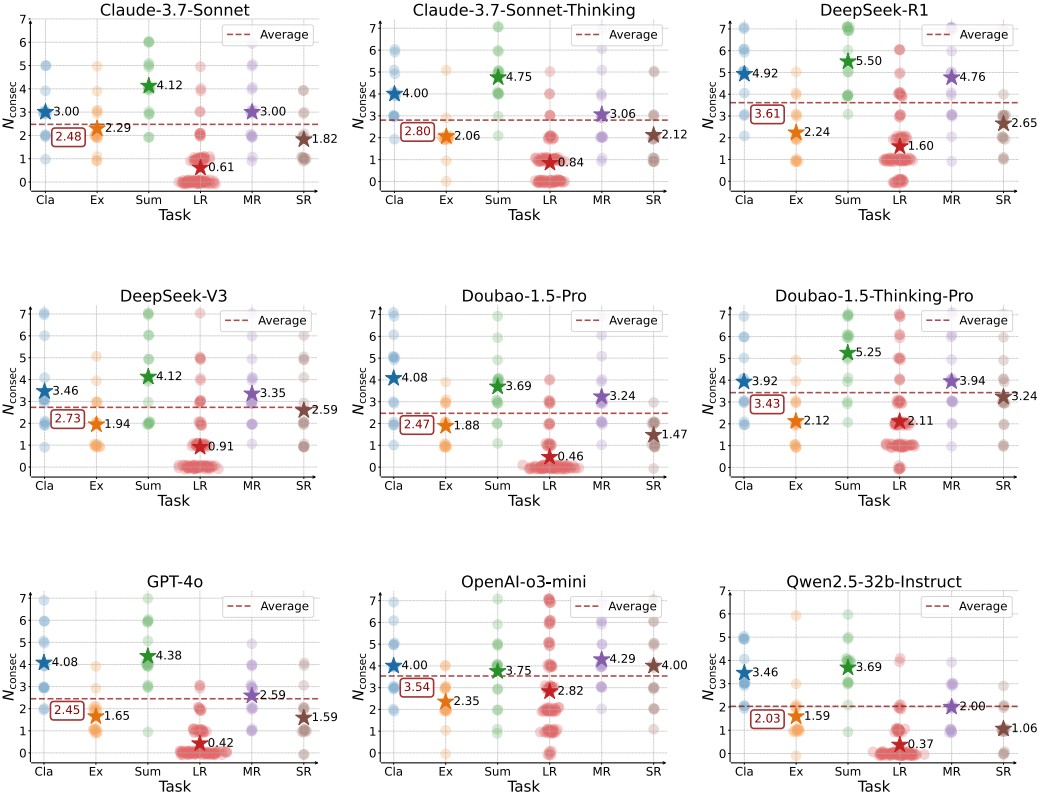

Figure 13: Results of the average number of consecutive correct solutions ($N_{\text{consec}}$) for demonstration learning.

## G.6 Results of Post-Warmup Accuracy ($\text{Acc}_{\text{pw}-K}$)

Figures 14 and 15 show the results of post-warmup accuracy ($\text{Acc}_{\text{pw}}$-K) for feedback learning and demonstration learning, respectively. This metric reflects model performance after an initial "warmup" phase, i.e., after some experience has been accumulated.

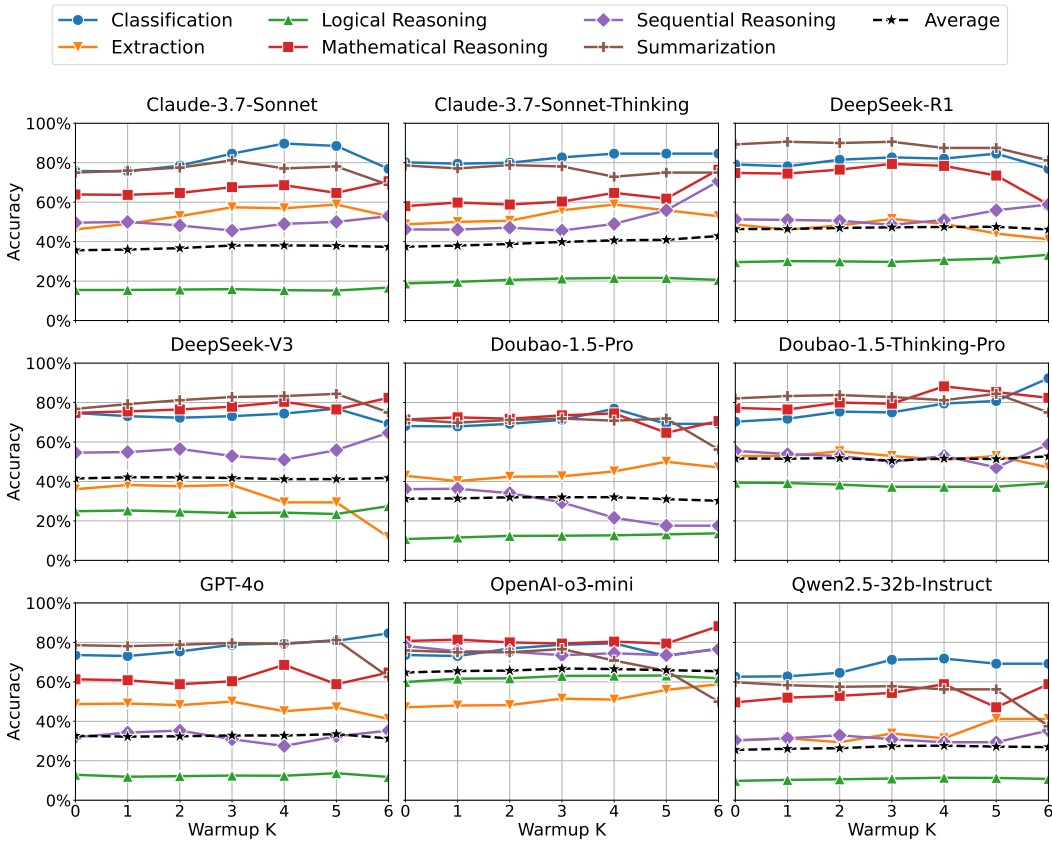

Figure 14: Results of post-warmup accuracy ($\text{Acc}_{\text{pw}}$-K) for feedback learning.

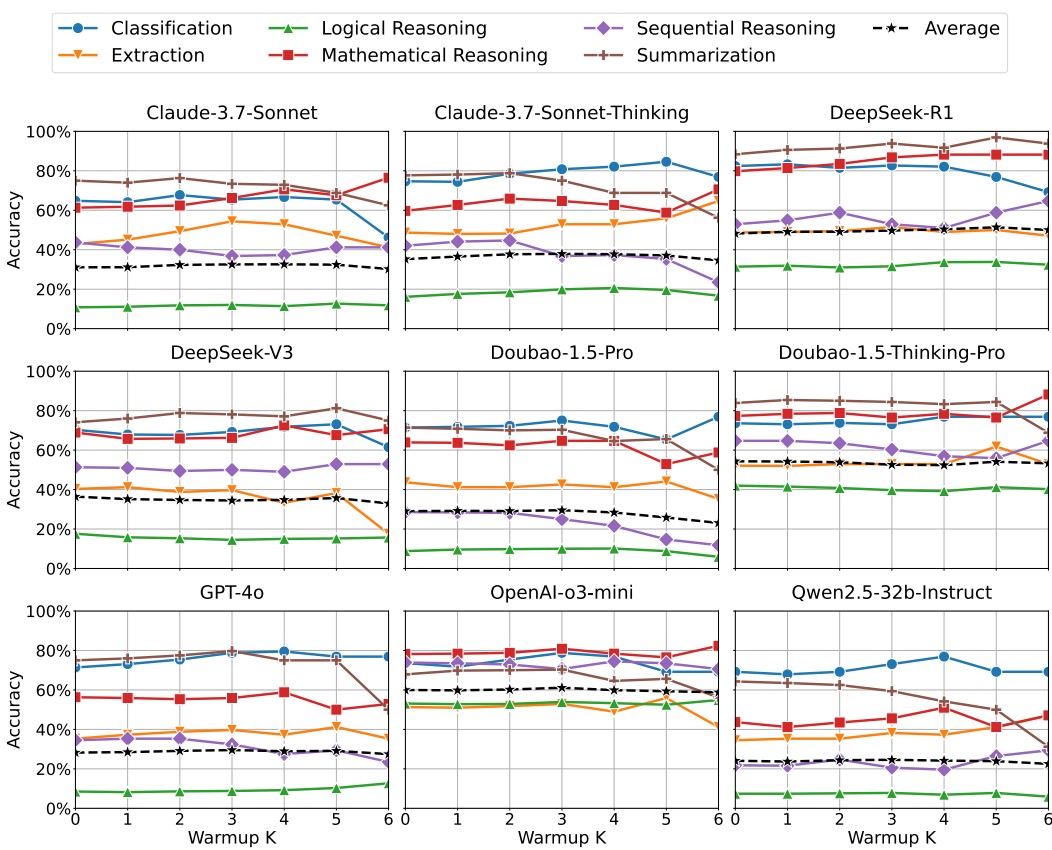

Figure 15: Results of post-warmup accuracy ($\mathrm{Acc_{pw}}$-K) for demonstration learning.

## G.7   Results of Position-wise Accuracy Curve

Figures 16 and 17 show the position-wise accuracy curves for feedback learning and demonstration learning, respectively. These curves display the average accuracy at each position across all sequences.

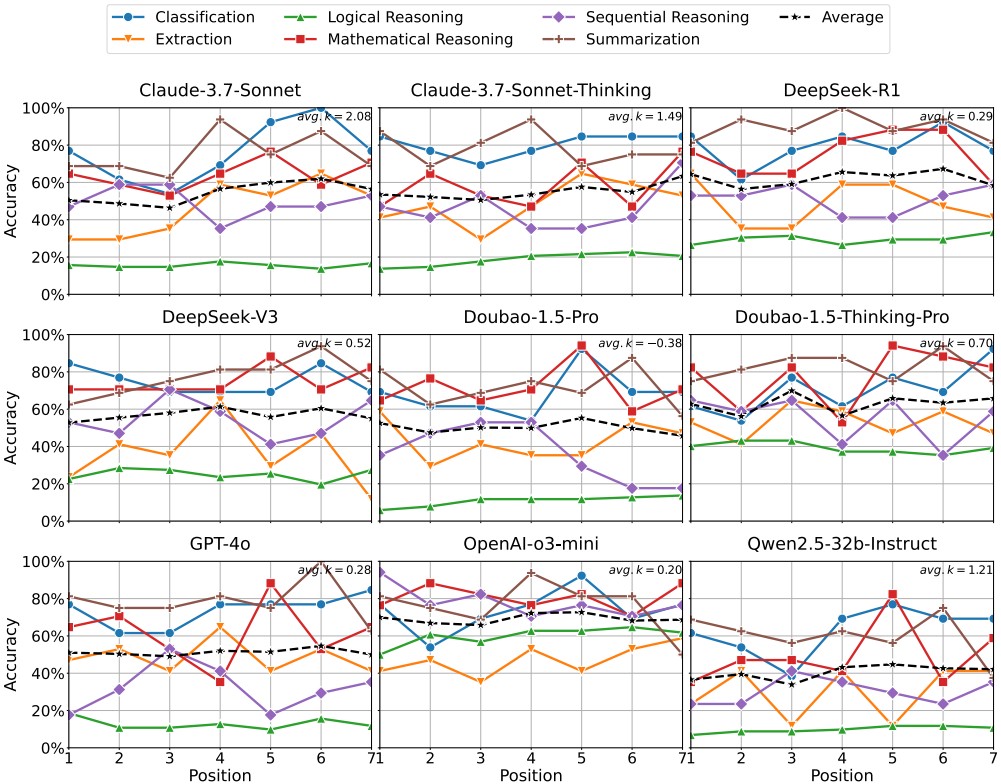

Figure 16: Position-wise Accuracy Curves for feedback learning.

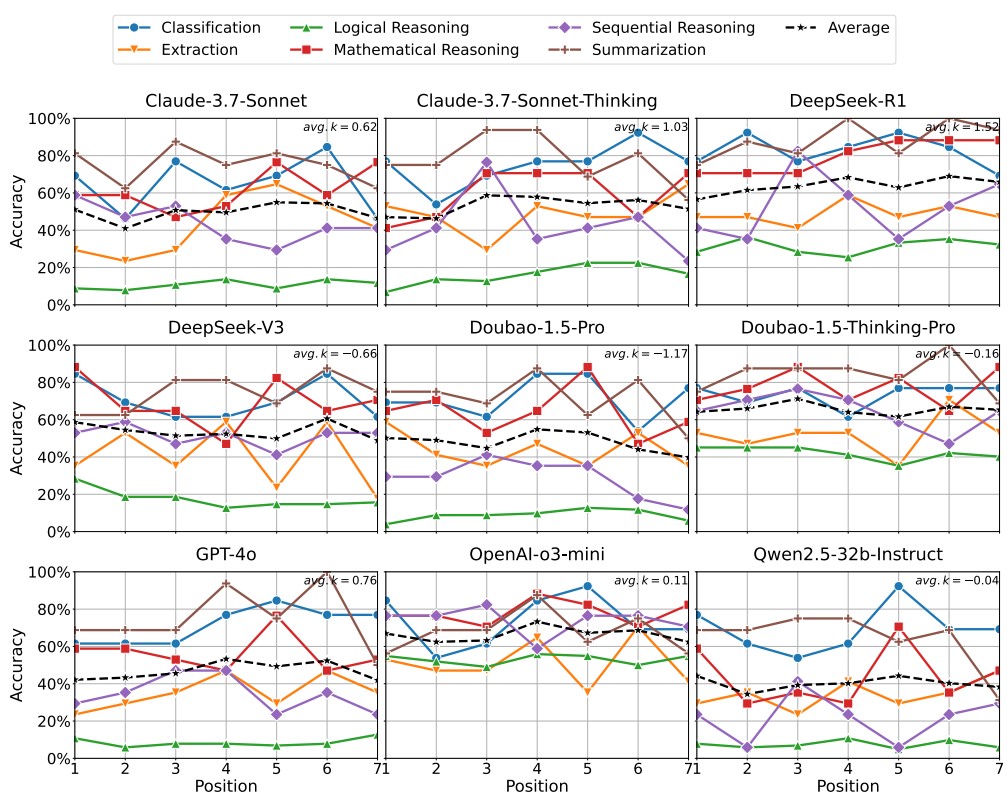

Figure 17: Position-wise Accuracy Curves for demonstration learning.

# H System Prompts

## H.1 Evaluation Prompt for LLM-as-a-judge

Figure 18 presents the evaluation prompt used in LLM-as-a-judge.

---

**Evaluation prompt for LLM-as-a-judge**

From now on, your role is a rigorous instruction-following grading teacher. Your task is to grade the student's answer strictly according to the standard answer. You need to follow the steps below for grading. This is very important to me. Before you begin, please note the following two points:

1. You have a two-level grading scale: 0 points and 1 point.0 points means the student's answer does not meet all the requirements in the standard answer. Each requirement in the standard answer is equally important. If even one requirement is not met, the score must be 0 points.1 point means the student's answer fully meets all the requirements in the standard answer.

2. When you're ready to begin grading, remain calm and focused. Analyze and think through the problem step by step, following these steps:

   - Carefully read and understand each requirement in the standard answer.
   - Analyze whether the student's answer fully follows all the requirements in the standard answer, comparing each part of the student's answer with the standard answer.
   - Do not rush to a conclusion. Before finalizing your grading, perform a Self-Reflection on the analysis: Ensure that your grading criteria take into account all the requirements of the standard answer and that no requirement is overlooked just because it seems "unimportant." Verify that your grading criteria and the score are logical and consistent. If there are any errors or omissions, correct them on time.
   - Once you're confident your analysis is correct, assign the grade based on your analysis and display it in the following "JSON" format. Be strict in adhering to the output format requirements.

**Output Format:** Your output should follow this exact format:
[Grading Rationale]:
[Score]: x points
[JSON]: {"answer_score": score}
**Example 1**
<Standard Answer>: The student's answer must include an emoji after the word "jump rope".
<Student's Answer>: Jump rope is an effective aerobic exercise that can help you burn calories. However, jump rope for weight loss requires consistent effort over time, and should be combined with a proper diet and other forms of exercise. If you want to lose weight through jump rope, it's recommended to do it for at least 30 minutes per day and gradually increase the difficulty and intensity. Also, watch your diet and avoid high-calorie, high-fat, high-sugar foods.
[Grading Rationale]: The word "jump rope" is not followed by an emoji.
[Score]: 0 points
[JSON]: {"answer_score": 0}
**Example 2**
<Standard Answer>: The student's answer must describe Beijing using a mix of Chinese and Korean.
<Student's Answer>: 北京啊，北京是中国的首都，是中国的政治中心、文化中心、国际交往中心、科技创新中心。北京有着悠久的历史和丰富的文化遗产，如故宫、长城、颐和园等。北京还是中国的经济中心之一，拥有众多的跨国公司和金融机构。北京是一个充满活力和机遇的城市，吸引着来自世界各地的人们前来旅游、学习和工作。
[Grading Rationale]: The student's answer uses only Chinese and does not include any Korean as required.

[Score]: 0 points
[JSON]: {"answer_score": 0}
**Example 3**
<Standard Answer>: The student's answer must ask about the user's needs.
<Student's Answer>:Can you tell me what problem you're facing? That way, I can answer more accurately
[Grading Rationale]: The student asked about the user's needs, meeting all the requirements of the standard answer.
[Score]: 1 point
[JSON]:{"answer_score": 1}
**Final Note:** I hope you will be able to take on the role of a grading teacher effectively, as this is very important to my work. If you perform well, I will reward you appropriately. Otherwise, I may impose some penalties. Here is the official question:

Figure 18: The evaluation prompt used in LLM-as-a-judge.

## H.2 System Prompts for Different Solving Methods

Prompt for Zero-shot

You are a student, you need to complete a question about **{type}** ability.
[Question]:**{question}**

Figure 19: System prompt for zero-shot (parallel solving).

Prompt for Few-shot

You are a student, you need to complete a question about **{type}** ability.
Before giving you the final question, I will provide some examples of other questions and their standard answers.
[History 1 Start]
[Question Start]
**{example_question_1}**
[Question End]
[Standard Answer Start]
**{example_answer_1}**
[Standard Answer End]
[History 1 End]
...
All history records have ended, now please begin answering the final question.
[Question]:**{question}**

Figure 20: System prompt for few-shot (parallel solving).

Prompt for Demonstration Learning

You are a student, you need to complete a question about **{type}** ability.
Before giving you the final question, I will provide some examples of other questions and their standard answers.
[History 1 Start]
[Question Start]
**{previous_question_1}**
[Question End]

[Standard Answer Start]
**{previous_answer_1}**
[Standard Answer End]
[History 1 End]
...
All history records have ended, now please begin answering the final question.
[Question]:**{question}**

Figure 21: System prompt for demonstration learning (sequential solving).

You are a student, you need to complete a question about **{type}** ability.
Before giving you the final question, I will provide some examples of other questions and their standard answers.
[History 1 Start]
[Question Start]
**{example_question_1}**
[Question End]
[Answer Evaluation Criteria Start]
**{previous_rubric_1}**
[Answer Evaluation Criteria End]
[Teacher's Evaluation of Student's Answer Start]
**{previous_judge_1}**
[Teacher's Evaluation of Student's Answer End]
[History 1 End]
...
All history records have ended, now please begin answering the final question.
[Question]:**{question}**

Figure 22: System prompt for feedback learning (sequential solving).

# I  Instances

Figures 23 and 24 showcase two instances in the logical reasoning task of EvaLearn.

The instance in Figure 23 involves solving a $4 \times 4$ sliding puzzle. In this case, the puzzle consists of 16 tiles, one of which is a blank space (denoted by '>'), while the remaining tiles are filled with different symbols. By moving the blank space and swapping it with adjacent tiles, the goal is to restore the puzzle from a given initial state to a target configuration through a series of logical moves. This task is designed to assess the model's logical reasoning ability, and all problems within this sequence are centered around the sliding puzzles and logical reasoning.

Figure 24 presents another problem from the same sequence, which differs mainly in grid size, the initial state of the puzzle, and symbol representation. These problems are interrelated and collectively challenge the model's logical reasoning skills. Humans are able to gain experience and improve their performance by solving a series of such problems. By requiring models to solve these problems sequentially within a sequence, EvaLearn evaluates whether models can similarly learn from experience, thereby measuring their learning efficiency.

Instance 1 in the Logic Reasoning task

**Problem Unique Id**: 557
**Task Type**: Logical Reasoning
**Prompt**:

You are a virtual expert in solving a $4\times4$ sliding puzzle problem. Please follow the instructions and rules below to complete the solution. Your goal is to reach the goal state with valid moves.

[The goal state]

```
{   D   >   -
q   ]   _   *
Z   n   N   C
P   S   |   E
```

[Instructions]

The puzzle consists of a $4 \times 4$ grid containing various symbols, with '>'representing the movable space. Only '>'can be moved horizontally or vertically, and the objective is to reach the goal state from a given initial state.

[Rules]

1. Only '>'can be moved horizontally or vertically.
2. Each move is chosen from the following set of options:

- 'Left': move '>'to the left;
- 'Down': move '>'downward;
- 'Right': move '>'to the right;
- 'Up': move '>'upward

3. The next move must be chosen from the valid move set depending on the position of '>'.
For example:

```
p1    p2    p3    p4
p5    p6    p7    p8
p9    p10   p11   p12
p13   p14   p15   p16
```

- If '>'is located at position 'p1', the valid move set is ['Down', 'Right'];
- If '>'is located at position 'p2', the valid move set is ['Down', 'Left', 'Right'];
- If '> 'is located at position 'p3 ', the valid move set is ['Down ', 'Left ', 'Right '];
- If '> 'is located at position 'p4 ', the valid move set is ['Down ', 'Left '];
- If '> 'is located at position 'p5 ', the valid move set is ['Up ', 'Down ', 'Right '];
- If '> 'is located at position 'p6 ', the valid move set is ['Up ', 'Down ', 'Left ', 'Right '];
- If '> 'is located at position 'p7 ', the valid move set is ['Up ', 'Down ', 'Left ', 'Right '];
- If '> 'is located at position 'p8 ', the valid move set is ['Up ', 'Down ', 'Left '];
- If '> 'is located at position 'p9 ', the valid move set is ['Up ', 'Down ', 'Right '];
- If '> 'is located at position 'p10 ', the valid move set is ['Up ', 'Down ', 'Left ', 'Right '];
- If '> 'is located at position 'p11 ', the valid move set is ['Up ', 'Down ', 'Left ', 'Right '];
- If '> 'is located at position 'p12 ', the valid move set is ['Up ', 'Down ', 'Left '];
- If '> 'is located at position 'p13 ', the valid move set is ['Up ', 'Right '];
- If '> 'is located at position 'p14 ', the valid move set is ['Up ', 'Left ', 'Right '];
- If '> 'is located at position 'p15 ', the valid move set is ['Up ', 'Left ', 'Right '];
- If '> 'is located at position 'p16 ', the valid move set is ['Up ', 'Left ']

4. Diagonal moves are not allowed.
5. The objective is to return the moves which can reach the goal state.

Please solve the following problem. Your answer should be in the format of a list of moves, for example:['Left ', 'Up ', 'Up '].
This problem can be solved within 1 to 12 steps. The next move must be chosen from the valid move set ['Left ', 'Right ', 'Up ', 'Down ']. The maximum step number you can take is 12. Try to reach the goal state using the least number of steps ($\leq 12$) ). DO NOT exceed 12 steps.
[Initial State]

```
q   {   _   -
Z   D   N   *
P   ]   |   C
>   n   S   E
```
**Canonical Answer**:
[ 'Up ', 'Up ', 'Up ', 'Right ', 'Down ', 'Down ', 'Down ', 'Right ', 'Up ', 'Up ', 'Up ']
**Rubric**:
The student's final answer must exactly match the content and order of the standard answer. The reasoning process, letter case, extra spaces, and other formatting issues can be ignored, but the final answer must contain all the correct elements of the standard answer.
Standard answer:
[ 'Up ', 'Up ', 'Up ', 'Right ', 'Down ', 'Down ', 'Down ', 'Right ', 'Up ', 'Up ', 'Up ']
If the student's answer differs in content from the standard answer (after ignoring case and formatting), the score is 0.
**Sequence Id**: 51
**Position In Sequence**: 1

Figure 23: Instance 1 in the logical reasoning task of EvaLearn.

Instance 2 in the Logic Reasoning task

**Problem Unique Id**: 557
**Task Type**: Logical Reasoning
**Prompt**:
You are a virtual expert in solving a $6 \times 6$ sliding puzzle problem. Please follow the instructions and rules below to complete the solving. Your goal is to reach the goal state with valid moves.
[The goal state]

```
a   9   ~   }   [   e
o   8   i   E   S   -
!   +   w   l   P   =
q   B   0   6   n   F
7   <   3   $   K   A
Y   u   &   p   O   U
```

[Instructions]
The puzzle consists of a $6 \times 6$ grid containing various symbols, with 'B' representing the movable space. Only 'B' can be moved horizontally or vertically, and the objective is to reach the goal state from a given initial state.

[Rules]
1. Only 'B' can be moved horizontally or vertically.
2. Each move is chosen from the following set of options:

- 'Left': move 'B'to the left;
- 'Down': move 'B'downward;
- 'Right': move 'B'to the right;
- 'Up': move 'B'upward

3. The next move must be chosen from the valid move set depending on the position of 'B'.
For example:

```
p1     p2     p3     p4     p5     p6
p7     p8     p9     p10    p11    p12
p13    p14    p15    p16    p17    p18
p19    p20    p21    p22    p23    p24
p25    p26    p27    p28    p29    p30
p31    p32    p33    p34    p35    p36
```
- If 'B' is located at position 'p1', the valid move set is ['Down', 'Right'];
- If 'B' is located at position 'p2', the valid move set is ['Down', 'Left', 'Right'];
- If 'B' is located at position 'p3', the valid move set is ['Down', 'Left', 'Right'];
- If 'B' is located at position 'p4', the valid move set is ['Down', 'Left', 'Right'];
- If 'B' is located at position 'p5', the valid move set is ['Down', 'Left', 'Right'];
- If 'B' is located at position 'p6', the valid move set is ['Down', 'Left'];
- If 'B' is located at position 'p7', the valid move set is ['Up', 'Down', 'Right'];
- If 'B' is located at position 'p8', the valid move set is ['Up', 'Down', 'Left', 'Right'];
- If 'B' is located at position 'p9', the valid move set is ['Up', 'Down', 'Left', 'Right'];
- If 'B' is located at position 'p10', the valid move set is ['Up', 'Down', 'Left', 'Right'];
- If 'B' is located at position 'p11', the valid move set is ['Up', 'Down', 'Left', 'Right'];
- If 'B' is located at position 'p12', the valid move set is ['Up', 'Down', 'Left'];
- If 'B' is located at position 'p13', the valid move set is ['Up', 'Down', 'Right'];
- If 'B' is located at position 'p14', the valid move set is ['Up', 'Down', 'Left', 'Right'];
- If 'B' is located at position 'p15', the valid move set is ['Up', 'Down', 'Left', 'Right'];
- If 'B' is located at position 'p16', the valid move set is ['Up', 'Down', 'Left', 'Right'];
- If 'B' is located at position 'p17', the valid move set is ['Up', 'Down', 'Left', 'Right'];
- If 'B' is located at position 'p18', the valid move set is ['Up', 'Down', 'Left'];
- If 'B' is located at position 'p19', the valid move set is ['Up', 'Down', 'Right'];
- If 'B' is located at position 'p20', the valid move set is ['Up', 'Down', 'Left', 'Right'];
- If 'B' is located at position 'p21', the valid move set is ['Up', 'Down', 'Left', 'Right'];
- If 'B' is located at position 'p22', the valid move set is ['Up', 'Down', 'Left', 'Right'];
- If 'B' is located at position 'p23', the valid move set is ['Up', 'Down', 'Left', 'Right'];
- If 'B' is located at position 'p24', the valid move set is ['Up', 'Down', 'Left'];
- If 'B' is located at position 'p25', the valid move set is ['Up', 'Down', 'Right'];
- If 'B' is located at position 'p26', the valid move set is ['Up', 'Down', 'Left', 'Right'];
- If 'B' is located at position 'p27', the valid move set is ['Up', 'Down', 'Left', 'Right'];
- If 'B' is located at position 'p28', the valid move set is ['Up', 'Down', 'Left', 'Right'];
- If 'B' is located at position 'p29', the valid move set is ['Up', 'Down', 'Left', 'Right'];
- If 'B' is located at position 'p30', the valid move set is ['Up', 'Down', 'Left'];
- If 'B' is located at position 'p31', the valid move set is ['Up', 'Right'];
- If 'B' is located at position 'p32', the valid move set is ['Up', 'Left', 'Right'];
- If 'B' is located at position 'p33', the valid move set is ['Up', 'Left', 'Right'];
- If 'B' is located at position 'p34', the valid move set is ['Up', 'Left', 'Right'];
- If 'B' is located at position 'p35', the valid move set is ['Up', 'Left', 'Right'];
- If 'B' is located at position 'p36', the valid move set is ['Up', 'Left'];

4. Diagonal moves are not allowed.
5. The objective is to return the moves which can reach the goal state. Please solve the following problem. Your answer should be in the format of a list of moves, for example: ['Left', 'Up', 'Up'].
This problem can be solved within 1 to 18 steps. The next move must be chosen from the valid move set ['Left', 'Right', 'Up', 'Down']. The maximum step number you can take is 18. Try to reach the goal state using the least number of steps (≤ 18). DO NOT exceed 18 steps.
[Initial State]

```
o   B   ~   }   [   e
!   a   i   E   S   -
q   9   w   l   P   =
7   8   0   6   n   F
Y   +   3   $   K   A
u   <   &   p   O   U
```

**Canonical Answer**:
['Down', 'Down', 'Down', 'Down', 'Down', 'Left', 'Up', 'Up', 'Up',
'Up', 'Up', 'Right', 'Down', 'Down', 'Down']
**Rubric**: The student's final answer must exactly match the content and order of the standard
answer. The reasoning process, letter case, extra spaces, and other formatting issues can be
ignored, but the final answer must contain all the correct elements of the standard answer.
Standard answer:
['Down', 'Down', 'Down', 'Down', 'Down', 'Left', 'Up', 'Up', 'Up',
'Up', 'Up', 'Right', 'Down', 'Down', 'Down']
If the student's answer differs in content from the standard answer (after ignoring case and
formatting), the score is $0$.
**Sequence Id**: $51$
**Position In Sequence**: $6$

Figure 24: Instance 2 in the logical reasoning task of EvaLearn. This instance and Figure 23 are from
the same sequence. The two problems differ primarily in grid size, the initial state of the puzzle, and
symbol representation.

