# OpenReview forum: "EvaLearn: Quantifying the Learning Capability and Efficiency of LLMs via Sequential Problem Solving"
_NeurIPS.cc/2025/Conference — NeurIPS 2025 poster_

### Official Review · Reviewer_djCX · 2025-06-27

**Clarity:** 4
**Significance:** 4
**Originality:** 4
**Rating:** 5
**Confidence:** 4

**Summary:**

This paper introduces EvaLearn, a novel benchmark for evaluating large language models' learning capability and efficiency. Unlike existing parallel evaluations that test models on i.i.d. samples, EvaLearn uses sequential problem-solving where models can learn and adapt within a given task sequence. The benchmark contains 648 problems across 182 sequences spanning six task categories. This work addresses a key gap in current evaluation by measuring LLMs' learning ability rather than static ability.

**Questions:**

- The paper presents a comprehensive benchmark of nine frontier LLMs, but there's a notable omission of any model from Google Gemini.
- The author mentions that the metric "(3) Average position of first correct solution" is proposed to measure "initial learning speed". However, a model with superior static capabilities might solve the first problem in a sequence correctly simply due to its pre-existing knowledge, without any "learning" taking place.

**Ethical Concerns:**

["NO or VERY MINOR ethics concerns only"]

**Final Justification:**

The author provides an extra set of experiments on the Google Gemini family. They also provide clear and detailed justifications on the proposed rubrics based on empirical results. My concerns are well addressed.

**Limitations:**

yes

**Quality:**

4

**Strengths And Weaknesses:**

Strengths
- The paper is well-written and organized.
- This work presents a novel benchmark to evaluate LLMs' learning ability, which remains unexplored in existing evaluations, and I believe it is of high significance to the research community.
- The whole benchmark is of very high quality, with carefully curated data, well-grounded evaluation methods, and sophisticated and meaningful rubrics.

Weaknesses
- There are no obvious weaknesses.

---

> ### Author Rebuttal · Authors · 2025-07-30
>
> Thank you very much for your recognition of our work and for providing valuable suggestions! We have addressed each of the questions you raised accordingly. We will incorporate all the new experiments and revisions into the next version of our manuscript.
>
> > **Question 1: The paper presents a comprehensive benchmark of nine frontier LLMs, but there's a notable omission of any model from Google Gemini.**
>
> **Answer:** Thank you for your valuable comments! Google Gemini has indeed demonstrated impressive performance in practical applications. To comprehensively investigate the dynamic learning potential of current frontier LLMs, we evaluated **two Gemini series models (i.e., Gemini-2.5-Flash and Gemini-2.5-Pro)** on EvaLearn. Gemini-2.5-Flash is Google’s frontier LLM, exhibiting strong capabilities across various tasks. Gemini-2.5-Pro is Google’s most advanced reasoning model, capable of solving more complex problems. The overall accuracies of the two LLMs under both the zero-shot and feedback learning settings are shown in the table below.
>
> |Model|Paradigm|Sum (%)|Cla (%)|Ex (%)|LR (%)|MR (%)|SR (%)|Avg. (%)|
> |----|----|---|---|---|---|---|---|----|
> |Gemini-2.5-Pro|Zero-shot|85.0|72.9|55.0|62.5|90.0|76.7|68.5|
> ||Feedback Learning|89.3 (+4.3)|75.8 (+2.9)|52.1 (-2.9)|59.9 (-2.6)|85.7 (-4.3)|79.8 (+3.1)|67.2 (-1.3)|
> |Gemini-2.5-Flash|Zero-shot|81.7|66.7|38.3|16.1|70.0|51.7|36.3|
> ||Feedback Learning|67.9 (-13.8)|59.3 (-7.4)|42.0 (+3.7)|18.3 (+2.2)|81.5 (+11.5)|58.8 (+7.1)|37.5 (+1.2)|
>
> From the table, we can draw several findings:
> (1) Compared to other frontier LLMs, Gemini-2.5-Pro demonstrates strong overall static performance, particularly in math reasoning and sequential reasoning tasks, with overall accuracy comparable to other thinking-based LLMs such as OpenAI-o3-mini. In contrast, Gemini-2.5-Flash’s overall performance is lower than Gemini-2.5-Pro, with results on some tasks similar to non-thinking-based models like Claude-3.7-Sonnet and GPT-4o.
> (2) For different tasks, both models exhibit effective sequential learning capabilities, which can leverage historical experience to solve new problems. Specifically, Gemini-2.5-Flash shows sequential learning capability in mathematical reasoning; for example, through feedback learning, its accuracy on math reasoning tasks increases from 70.0 to 81.5. Gemini-2.5-Pro excels at utilizing experience and feedback to solve new problems in tasks such as summarization and classification.
>
> We further analyze the performance of these two models on other metrics to explore their sequential learning potential in greater depth. The results are shown in the table below. QoQ denotes the quarter-on-quarter growth rate.
>
> | Metric|Model| Cla | Ex | LR | MR | SQ | Sum | Avg.|
> |------|------|-----|-----|-----|-----|----|------|---|
> | QoQ| Gemini-2.5-Flash| 3.37%| 3.98%| 9.27%| 2.23%| 2.38%| -2.89%| -0.59%|
> | QoQ| Gemini-2.5-Pro| 2.68%|6.22%| -0.8%| 3.66%| 0.96%| -3.18%| 0.45%|
> | P_first  | Gemini-2.5-Flash  | 3.0 | 2.76| 6.55 | 3.4   | 3.18  | 4.08 | 3.83 |
> | P_first  | Gemini-2.5-Pro | 3.0| 3.47| 3.66| 2.71  | 3.46  | 4.31  | 3.44|
> | N_consec | Gemini-2.5-Flash | 2.77| 1.82| 1.07 | 4.82 | 2.41  | 4.06  | 2.83 |
> | N_consec | Gemini-2.5-Pro  | 4.15  | 2.41 | 3.35 | 5.06 | 4.65  | 5.81 | 4.24  |
>
> From the table, we can further observe the following phenomena:
>
> (1) In terms of QoQ, Gemini-2.5-Pro performs relatively better; for example, its improvement rate on extraction tasks is 6.22%, while Gemini-2.5-Flash achieves 3.98%. According to this metric, these two models show moderate learning efficiency compared to the nine previously evaluated models.
>
> (2) On the new $P_\text{first}$ metric (as shown in the response to Question 2), Gemini-2.5-Pro demonstrates excellent performance, second only to OpenAI-o3-mini, whereas Gemini-2.5-Flash ranks at a moderate level.
>
> (3) Additionally, for the $N_\text{consec}$ metric, Gemini-2.5-Pro achieves the best performance among all models, indicating superior learning stability and a stronger ability to leverage prior experience to solve new problems.
>
> (4) After incorporating the two newly added Gemini series LLMs, all the general conclusions in our manuscript remain unchanged. Specifically, models that perform well in terms of static capability on certain tasks do not necessarily exhibit strong dynamic learning abilities. The dynamic learning potential of models still requires further attention from the research community.
>
> Overall, Gemini-2.5-Pro shows **impressive performance** in sequential learning.
> Gemini-2.5-Flash also shows good learning ability on certain tasks; however, it does not exhibit a clear advantage over these thinking-based LLMs.
>
> We sincerely thank the reviewer for this valuable suggestion. We will further visualize the results of these two models across all metrics and include them in the latest version of our manuscript. We believe this will help readers gain a deeper understanding of this overlooked sequential learning abilities of current mainstream LLMs.
>
> > **Question 2: A model with superior static capabilities might solve the first problem in a sequence correctly simply due to its pre-existing knowledge, without any "learning" taking place.**
>
> **Answer:** Thank you for your valuable comments. You are right—the $P_\text{first}$ metric reflects both the model’s static performance and its dynamic sequential learning efficiency. A model with superior static capabilities might solve the first problem in a sequence correctly simply due to its pre-existing knowledge.
>
> To address this, we conduct additional experiments to **decouple these two aspects within this metric,** allowing us to further investigate the model’s learning efficiency. Specifically, for each sequence, we first identify the position of the first problem that the model answers incorrectly in the zero-shot setting. This position serves as the starting point for feedback-based learning. Then, when calculating the "position of first correct solution" for this sequence, we exclude those problems that the model had already answered correctly in the zero-shot setting.
>
> By doing so, we ensure that the calculation of the ​$P_\text{first}$ metric excludes problems solved due to pre-existing knowledge, thus providing a more accurate measure of the model’s learning efficiency. Under this condition, $P_\text{first}$ can be formulated as follows:
>
> $$
> P_{\text{first}} = \min (i - i_0),\ \text{for } i \geq i_0,\ A^{\text{feedback}}_i = 1,\ A^{\text{zero}}_i = 0
> $$
>
> where $ i_0 $ denotes the position of the first problem that the model answers incorrectly in the zero-shot setting. $ A_i $ is a binary indicator representing whether the model's answer at position $ i $ is correct (1) or incorrect (0).
> The experimental results regarding $P_\text{first}$ are shown in the below table:
>
> |Model|Cla|Ex|Sum|LR|MR|SR|Avg.|
> |---|---|---|---|---|---|---|---|
> |Doubao-1.5-Pro|3.58|3.71|3.54|6.75|3.24|3.06|3.98|
> |DeepSeek-V3|4.0|3.88|3.82|5.27|3.27|4.00|4.04|
> |DeepSeek-R1|4.5|2.76|4.7|4.7|3.62|4.19|4.08|
> |Gemini-2.5-Flash|3.0|2.76|4.08|6.55|3.4|3.18|3.83|
> |Claude-3.7-Sonnet|3.08|3.71|4.08|5.69|3.24|1.94|3.62|
> |Gemini-2.5-Pro|3.0|3.47|4.31|3.66|2.71|3.46|3.44|
> |Claude-3.7-Sonnet-Thinking|2.77|3.18|3.8|5.55|3.41|2.53|3.54|
> |GPT-4o|3.45|3.24|4.08|6.0|2.41|3.29|3.75|
> |Qwen2.5-32b-Instruct|3.69|3.35|3.93|6.38|3.65|3.53|4.09|
> |Doubao-1.5-Thinking-Pro|3.83|2.88|4.9|3.44|3.67|4.81|3.92|
> |OpenAI-o3-mini|3.42|3.71|2.6|2.93|3.4|3.62|3.28|
>
> From the table, we can draw the following conclusions:
>
> (1) After excluding problems that the LLM could already solve in the zero-shot setting, the average position of the first correct solution in sequential learning is relatively late for frontier LLMs. This indicates that these models face challenges in leveraging past problem-solving experiences to solve new problems.
>
> (2) There are differences in the $P_\text{first}$ across different task types. For example, LLMs generally achieve the first correct solution earlier in classification and extraction tasks compared to other tasks. This suggests that current LLMs may find it easier to summarize and utilize experience in these discriminative and extraction tasks.
>
> (3) The long chain-of-thought reasoning paradigm provides a slight improvement in the learning efficiency of models. Most thinking-based LLMs show slightly better learning efficiency compared to their non-thinking-based counterparts within the same series.
>
> (4) Finding 5 in our manuscript remains unchanged: The position of the first correct solution varies significantly across models and tasks, and does not align with overall accuracy. This highlights that EvaLearn assesses a novel aspect of models’ dynamic learning potential, distinct from their static capabilities. EvaLearn encourages the research community to further investigate the dynamic abilities of models, promoting the development of more powerful and human-like LLMs.
>
> In summary, the $P_\text{first}$ under the new conditions decouples the model’s static capability from its dynamic learning efficiency, allowing readers to better understand the dynamic learning potential of current LLMs. We will include these new experiments, findings, and further analyses in our latest manuscript to make the paper more comprehensive and robust.
>
>
> Finally, we sincerely appreciate the reviewer’s recognition of our work, careful review, and valuable suggestions. We will **incorporate all additional experiments and revisions** into our latest manuscript. If you have any further comments, please do let us know, and we will do our best to address them.

---

> > ### Comment · Reviewer_djCX · 2025-08-02
> >
> > Thank you for the additional experimental results and clarifications, which have addressed my questions.
> > I will maintain my score, as it was already high.

---

### Official Review · Reviewer_6LKa · 2025-06-30

**Clarity:** 3
**Significance:** 3
**Originality:** 3
**Rating:** 5
**Confidence:** 3

**Summary:**

The paper introduces EvaLearn, a new benchmark that probes dynamic learning in large language models (LLMs). Instead of the standard i.i.d. “parallel” tests, EvaLearn presents 182 short sequences (7 items each, 648 problems total) spanning six task families (summarization, classification, extraction, logical, mathematical and sequential reasoning). Models solve each sequence step-by-step, optionally receiving demonstrations or rubric-based feedback after every attempt. Five automated metrics (overall accuracy, slope of position-wise accuracy curve, position of first success, longest correct streak, and post-warm-up accuracy) quantify both learning capability and learning efficiency. Experiments with nine frontier models (thinking-based vs. non-thinking) reveal large variance: some models (e.g., Claude-3.7-Sonnet-Thinking, o3-mini) learn rapidly, others regress, and static leaderboard strength is only weakly correlated with sequential learning skill.

**Questions:**

1. Have you tried an ensemble of judges or self-consistency to mitigate single-LLM bias? Reporting Cohen’s κ between two judges would strengthen confidence.

2. Did you vary M (number of items per sequence)? Learning slopes might change if sequences are longer/shorter.

3. How were problem difficulties equalised across tasks? Presenting per-task baseline human or smaller-model performance would contextualise scores.

4. Any plan to add “in-the-wild” tasks (e.g., debugging a small codebase, multi-hop search) where learning from prior mistakes is crucial?

**Ethical Concerns:**

["NO or VERY MINOR ethics concerns only"]

**Final Justification:**

Thank you for your clarification and explaination. All my concerns are well addressed and I believe this is a good contribution. I will keep my original score.

**Limitations:**

The authors include a limitations section but should expand on data provenance and potential bias introduced by GPT-4o judging. No immediate societal-risk issues are apparent, but public release requires a clear license and privacy audit of any real-world text used.

**Quality:**

3

**Strengths And Weaknesses:**

Strengths

1. Shifts evaluation from static competence to within-task learning, a dimension mostly absent from prior LLM benchmarks or agent arenas.

2. Provides a concrete, reproducible suite plus automated rubric-plus-LLM judging (≈95 % agreement with humans) that could become a standard for meta-learning research.

3. Careful experimental design: 648 hand-crafted, rubric-verified items; comparison of zero-shot, few-shot, demonstration and feedback paradigms; results interpreted with clear statistics (e.g., slopes, warm-up curves).

4. Paper is well-structured, with motivating figures (Fig. 1), metric definitions (Eqns 1-6) and ablation findings (Tables 2/8, Fig. 2-4) that make replication straightforward.

Weaknesses

1. Ground-truth judging depends on GPT-4o. Even with 95 % spot-checks, subtle rubric-matching errors or model-to-model bias may skew results; no cross-LLM agreement study is provided.

2. Task set is still synthetic and modest (182 × 7). It is unclear whether the observed learning behaviours transfer to real-world domains (e.g., code, multimodal, open-ended dialogue).

3. Sequential evaluation echoes earlier interactive / curriculum work (DROP-hard, AgentBench, REFLEXION). The paper could more explicitly position itself against those and justify design choices (why exactly 7 problems per sequence?).

---

> ### Author Rebuttal · Authors · 2025-07-30
>
> Thank you very much for your recognition and valuable suggestions! We have addressed all weaknesses and questions you raised, and incorporated additional experiments accordingly. We will carefully revise our manuscript and include all new experiments in its next version.
>
> >**Weakness 1 & Question 1: Ground-truth judging depends on GPT-4o. Even with 95% spot-checks, subtle rubric-matching errors or model-to-model bias may skew results; no cross-LLM agreement study is provided. Have you tried an ensemble of judges or self-consistency to mitigate single-LLM bias? Reporting Cohen’s κ between two judges would strengthen confidence.**
>
> **Answer:**   Thank you for raising concerns regarding the potential bias of using a single LLM as a rubric-based judger. To address this, we conduct a comprehensive inter-model agreement analysis using three leading LLMs: GPT-4o, DeepSeek-V3, and Doubao-1.5-Pro. All models are evaluated on the full EvaLearn set. We compute both raw agreement rates and Cohen’s κ coefficients:
>
> |JudgePair|Both 0|Both 1|RawAgreement|Cohen’sκ|
> |---|---|---|---|---|
> |GPT-4o vs DeepSeek-V3|464|174|98.5%|0.961|
> |GPT-4o vs Doubao-1.5-Pro|463|176|98.6%|0.965|
> |DeepSeek-V3 vs Doubao-1.5-Pro|471|174|99.5%|0.988|
>
> Results show that all pairwise comparisons demonstrate extremely **high agreement:** raw agreement rates exceed 98%, and all Cohen’s κ values are above 0.96 (“almost perfect agreement” by standard interpretation). Disagreements are extremely rare and minor. Combined with the results of human validation presented in our manuscript, these findings confirm that the automated evaluation in EvaLearn is **highly reliable and effective.** We will include these results in the revised manuscript.
>
> >**Weakness 2 & Question 4: Task set is still synthetic and modest (182 × 7). It is unclear whether the observed learning behaviours transfer to real-world domains (e.g., code, multimodal, open-ended dialogue). Any plan to add “in-the-wild” tasks (e.g., debugging a small codebase, multi-hop search) where learning from prior mistakes is crucial?**
>
> **Answer:**   Thank you for your valuable suggestions! EvaLearn is the pioneering benchmark that proposes a novel sequential evaluation framework to assess the learning potential of LLMs. To ensure robustness, we use human-synthesized data with strict quality control (details in Appendix C). The number of problems in EvaLearn is comparable to other mainstream benchmarks (as shown in our response to question 1 of Reviewer 5iax). EvaLearn aims to draw the community’s attention to the important but previously underexplored dynamic ability of learning potential.
>
> We agree that a model’s dynamic learning potential may reflect its ability to solve problems in real-world domains. In coding scenarios, LLMs often need to solve a series of related issues within a code repository; in open-domain multi-turn dialogues, LLMs engage in conversations centered around a single topic. These situations require leveraging historical experience to solve new problems. In the future, we plan to expand the evaluation to “in-the-wild” scenarios such as repository-level and codebase-level code editing, as well as multi-turn tasks. We also intend to broaden our evaluation paradigm to the multimodal domain, and further explore the relationship between dynamic learning ability and a model’s problem-solving capacity in real-world domains.
>
> We hope EvaLearn will inspire more researchers to focus on models’ learning potential, expand evaluation objectives, and develop algorithms to enhance models’ dynamic learning abilities. We will incorporate these statements and future work into our manuscript.
>
> >**Weakness 3: Sequential evaluation echoes earlier interactive / curriculum work (DROP-hard, AgentBench, REFLEXION). The paper could more explicitly position itself against those and justify design choices (why exactly 7 problems per sequence?).**
>
> **Answer:**  Thank you for your suggestion! We have carefully reviewed the three related works you mentioned. DROP [1] is a reading comprehension dataset focused on discrete reasoning, which evaluates static reasoning abilities in a parallel way. AgentBench [2] assesses LLMs’ agentic performance in multi-task environments, but does not focus on dynamic evaluation of learning ability and efficiency. **EvaLearn is significantly different:** it sequentially evaluates models on related problems and uses multiple metrics to quantify learning potential, emphasizing dynamic model abilities. Reflexion [3] is an algorithmic framework that enables LLMs to reflect and improve their answers based on feedback.
>
> Regarding the choice of seven problems per sequence, this was based on two considerations: (1) Too few problems are insufficient to evaluate whether the model can leverage prior experience to solve subsequent problems. (2) Too many problems would decrease the total number of sequences, potentially introducing evaluation errors. After balancing sequence length and total number, we decided on seven problems per sequence.
>
> We will add a discussion of these related works and highlight the differences with EvaLearn, as well as the rationale behind EvaLearn’s design, in our latest manuscript.
>
> >**Question 2: Did you vary M (number of items per sequence)? Learning slopes might change if sequences are longer/shorter.**
>
> **Answer:**   Thank you for your suggestion! We calculate the coefficient of variation (CV) of the learning slopes for sequence lengths of 5, 6, and 7:
>
> |Model|CV|
> |---|---|
> |Doubao-1.5-Thinking-Pro|0.104|
> |Claude-3.7-Sonnet|0.136|
> |Qwen2.5-32b-Instruct|0.203|
> |Claude-3.7-Sonnet-Thinking|0.305|
> |DeepSeek-V3|0.330|
> |GPT-4o|0.493|
> |O3‑mini|0.474|
> |DeepSeek-R1|0.497|
> |Doubao-1.5-Pro|0.513|
>
> Most LLMs exhibit stability (CV < 0.5), with only Doubao-1.5-Pro showing slightly higher variability. Under the same evaluation conditions, the majority of models show low CVs, indicating EvaLearn is effective and reliable for evaluating models’ learning efficiency. We will include this experiment in our latest manuscript.
>
> >**Question 3: How were problem difficulties equalised across tasks? Presenting per-task baseline human or smaller-model performance would contextualise scores.**
>
> **Answer:**   Thank you for your suggestion. We evaluated the accuracy of Qwen2.5-3B-Instruct on EvaLearn:
>
> |Task|Sum (%)|Cla (%)|MR (%)|Ex (%)|SR (%)|LR (%)|
> |-|-|-|-|-|-|-|
> |Acc|28.3|22|16.7|11.7|6.7|4.7|
>
> Results show that the accuracy for all tasks is below 30%, and there are certain differences in model performance across tasks, reflecting varying levels of task difficulty for the models. However, even for tasks that models perceive as similarly difficult (i.e., tasks with comparable accuracy), the difficulty of constructing problems that can challenge the models is not necessarily the same for human annotators. This subjective difference in perceived difficulty makes it challenging to achieve perfectly balanced task difficulty across different tasks.
>
> In practice, we exercise strict control over the difficulty of each problem (ensuring that each question is sufficiently challenging, since current frontier LLMs possess very strong static capabilities. If the problem is not difficult enough, models can often solve it directly using pre-existing knowledge, which fails to effectively assess their learning and generalization abilities). The specific process for controlling problem difficulty is presented in Appendix C of our manuscript **due to space limitations in the main text.**
> In brief, we randomly select three models from the list of SOTA models (to avoid designing problems biased toward any particular model) and verify manually that none of these models could solve the problem correctly, ensuring the challenge level of each question.
>
> Overall, the new baseline scores from smaller LLMs not only provide an effective reference for comparing the performance of frontier LLMs, but also help reveal the actual difficulty of each task. We believe that readers can use these to more accurately understand the challenges posed by different tasks and the potential for model improvement, enabling a more comprehensive assessment of the models’ actual capabilities.
> Once again, we thank the reviewer for this very helpful suggestion. We will include all results and revisions in our latest manuscript to help readers better understand the learning potential of models across different tasks.
>
>
> >**Limitations: The authors include a limitations section but should expand on data provenance and potential bias introduced by GPT-4o judging.**
>
> **Answer:**  Thank you for your suggestion! We have conducted consistency experiments involving multiple LLMs as rubric-based judgers, as shown in our response to Weakness 1 & Question 1. The results demonstrate our automated evaluation framework is robust. We will discuss the potential limitations of this pipeline, specifically the uncertainty about whether GPT-4o as a judger remains consistently reliable when the evaluation is extended to other tasks or modalities, and incorporate these discussions into the revised manuscript.
>
> Thank you again for recognizing our work and providing valuable insights. We will incorporate all new experiments and revisions into the next version of our manuscript to inspire further research in this area. If you have any further questions or suggestions, please let us know.
>
> **References:**
> [1] Dua, D., Wang, Y., Dasigi, P., Stanovsky, G., Singh, S., & Gardner, M. (2019). DROP: A Reading Comprehension Benchmark Requiring Discrete Reasoning Over Paragraphs. In Proceedings of NAACL-HLT (pp. 2368-2378).
>
> [2] Liu, X., Yu, H., Zhang, H., Xu, Y., Lei, X., Lai, H., ... & Tang, J. (2024, January). AgentBench: Evaluating LLMs as Agents. In ICLR.
>
> [3] Shinn, N., Cassano, F., Gopinath, A., Narasimhan, qK., & Yao, S. (2023). Reflexion: Language agents with verbal reinforcement learning. Advances in Neural Information Processing Systems, 36, 8634-8652.

---

> > ### Comment · Area_Chair_HAfq · 2025-08-05
> > **Reminder: Reviewer 6LKa first response**
> >
> > Dear reviewer 6LKa,
> >
> > This is a reminder to post your first response, as the deadline of author-reviewer discussion period is closing. The authors have responded to your reviews, and also to the others reviews. Please discuss openly with the authors, regarding your reviews and the addressed questions from the authors.

---

### Official Review · Reviewer_xW5S · 2025-07-01

**Clarity:** 3
**Significance:** 3
**Originality:** 3
**Rating:** 5
**Confidence:** 4

**Summary:**

This paper introduces a new LLM benchmark EvaLearn. EvaLearn is different from other benchmarks in that it evaluates LLMs’ sequential learning capabilities. EvaLearn consists of sequences that incorporate multiple problems, and the LLM is instructed to solve the problems sequentially with LLM-as-a-Judge providing feedback to the previous problems. Therefore, an LLM with a strong learning capability should be able to learn to improve from the past feedback to solve the future problems better. Overall EvaLearn consists of 648 problems organized into 182 sequences, and covers six task types. The authors also conduct a wide range of analysis to understand the LLMs’ behavior on their benchmark.

**Questions:**

- How do you prompt the judge model to give feedback for Feedback learning? In the Appendix only the prompt for grading is provided.
- How did you construct the judge rubrics for the summarization task? Since unlike the rest of the task, summarization is relatively subjective.

**Writeup/Style**
- A bit more clarification on what the “Sequential Reasoning” task is would be nice (L:98).
- Sentence in L:109 (This task…) and L:113 (These problems…) seems a bit redundant.
- o3-mini is sometimes written with an upper case O and sometimes with a lower case o.

**Ethical Concerns:**

["NO or VERY MINOR ethics concerns only"]

**Final Justification:**

This paper provides a unique perspective on LLM benchmarks, specifically investigating how well an LLM can learn during test time.

While I had few concerns regarding the later part of the paper, they were mostly addressed by additional experiments by the authors or they were misunderstandings.

I believe this work is a good contribution.

**Limitations:**

Yes

**Paper Formatting Concerns:**

No formatting concerns

**Quality:**

3

**Strengths And Weaknesses:**

### **Strengths**

- The motivation and the novelty of the benchmark is good. Oftentimes LLM chat assistants will not get something right in the first turn of conversation, which requires users to provide feedback. Evaluating how well can LLMs incorporate such feedback is an important aspect that is not well covered by existing benchmarks.

- The metrics used by the paper are unique/interesting and fit the purpose of the benchmark well.

- The paper is for the most part easy to read albeit it does have some minor ambiguities.

### **Weaknesses**

**False “Yes” in the Paper Checklist (L:1044, Experiment statistics significance)**
- The authors do not report statistical significance of their experiments (which I believe is necessary for this work in two different cases as explained below), yet they put “Yes” in the checklist.

**Concerns about many of the “Findings”.**

1. **[Findings 1]** If I’m not mistaken the authors do not compare the performance of Feedback learning against Demonstration learning in the main sections, which I believe is necessary to claim that the LLMs truly learn from the LLM-as-a-Judge feedback (instead of solely from the few-shot demonstrations). Instead the authors compare Feedback learning against Zero-shot in Findings 1, which from a broader perspective is few-shot inference vs zero-shot inference, and in my opinion is not a fair setting.

2. **[Findings 2 and 5]** Were the seven problems for each sequence randomly selected/ordered (this is not clear from the paper)? In that case the variance caused by the randomness could be a considerable factor for all of the experiments. In particular, I believe any analysis regarding $N_{consec}$ and $P_{first}$ will be affected significantly (Findings 2 and Findings 5) as these metrics will deviate a lot on whether easier/harder problems are grouped together or appear early. This would’ve been less of an issue if the sample size was large, but the benchmark only has 182 sequences. The authors should’ve conducted multiple runs across different random compositions of the sequences and reported the standard deviations.

3. **[Findings 4]** The authors fit linear regression models and use the slope to claim that some LLMs achieve increasing accuracy with the number of demonstrations/feedbacks, but they do not report the statistical significance (e.g. p-values). Looking at Figure 4 and 17, in most cases there seems to be no noticeable correlation or trend in the graphs. It’s possible that in most cases the regression models are not statistically significant, therefore unable to support the authors’ claim.

4. **[Findings 6]** The authors claim that Demonstration learning yields better results than Few-shot parallel solving, but isn’t that expected since demonstration learning overall has more in-context learning examples? In 71% (5/7) of the cases, the demonstration learning will have equal or more in-context examples than the few-shot parallel learning, which is fixed to three examples.


While I believe that this paper provides an important and novel perspective to LLM benchmarks, the later half of the paper with the "Findings" has non-trivial concerns that the authors should address.

---

> ### Author Rebuttal · Authors · 2025-07-30
>
> Thank you very much for your valuable suggestions. We have carefully addressed each issue and incorporated additional experiments as recommended. All new experiments and revisions will be included in the next version of our manuscript.
>
> > **Weakness 1: False “Yes” in the Paper Checklist.**
>
> **Answer:** We sincerely apologize for the error in our Paper Checklist. Following your suggestion, we repeated the experiments **five times** with different random compositions and reported the confidence intervals (**CIs**), as detailed in our response to Weakness 3. This strengthens the validity of EvaLearn’s evaluation framework and experimental results. We will integrate CI calculations into our automated framework and ensure all checklist items are accurately reflected in the manuscript.
>
> > **Weakness 2: [Findings 1] If I’m not mistaken, the authors do not compare the performance of Feedback learning against Demonstration learning in the main sections, which I believe is necessary to claim that the LLMs truly learn from the LLM-as-a-Judge feedback.**
>
> **Answer:** We would like to clarify that our manuscript reports model performance under zero-shot, few-shot, demonstration, and feedback learning settings, with the relevant table currently in the Appendix (Table 4), **due to space limitations in the main text**. We observe that, for most models, feedback learning outperforms demonstration learning, indicating LLMs can indeed learn from LLM-as-a-Judge feedback. Additionally, in Findings 6, we compare these approaches across more metrics and find that feedback learning achieves higher average accuracy and learning efficiency for most LLMs. We will move this observation and Table 4 to the main text to improve clarity if the paper is accepted.
>
> > **Weakness 3: [Findings 2 and 5] Were the seven problems for each sequence randomly selected/ordered (this is not clear from the paper)? The authors should’ve conducted multiple runs across different random compositions of the sequences and reported the standard deviations.**
>
> **Answer:** We apologize for previously placing the statement about randomization in the appendix (Line 772). In EvaLearn, problems within each sequence are randomly ordered. **We will move this statement to the main text.** Following your advice, we conducted **five** runs with different random compositions and reported the 95% confidence intervals (CIs) based on standard deviation, as shown in the table.
>
> |Model|Acc|Acc 95% CI|$P_\text{first}$|$P_\text{first}$ 95% CI|Quarter-on-quarter growth rate (QoQ)|
> |---|---|---|---|---|---
> |Doubao-1.5-Thinking-Pro|53.4%|(51.8%,55.0%)|1.653|(1.433,1.873)|0.5%|
> |Claude-3.7-Sonnet|39.0%|(37.5%,40.6%)|2.124|(1.551,2.696)|-0.4%|
> |Deepseek-V3|43.9%|(42.4%,45.5%)|2.043|(2.025,2.062)|1%|
> |Doubao-1.5-Pro-256k|30.9%|(29.4%,32.4%)|2.620|(1.959,3.282)|1.1%|
> |GPT-4o|32.0%|(30.6%,33.5%)|2.645|(1.856,3.433)|2.4%|
> |Deepseek-R1|48.2%|(46.6%,49.8%)|1.681|(1.412,1.951)|-0.1%|
> |Claude-3.7-Sonnet-Thinking|37.7%|(36.2%,39.2%)|2.018|(1.809,2.227)|3.4%|
> |Qwen2.5-32b-Instruct|26.5%|(25.1%,27.9%)|2.951|(2.071,3.830)|4.0%|
>
> The results show that all models have narrow CIs for accuracy (all less than 0.04), indicating stable performance across different random tests. For the $P_\text{first}$ metric, the width of the CIs varies among models, but overall, repeated experiments have significantly improved the credibility of the results and provided a more accurate assessment of the models’ true performance.  We also compared the new  $P_\text{first}$ rankings with those in our manuscript and found only one LLM’s ranking changed. We will incorporate all new experiments and carefully revise our manuscript to update our conclusions accordingly, ensuring rigor and reliability.
>
> Furthermore, we will **integrate the CI calculation code into the EvaLearn evaluation framework** and open-source it together with the new sequences to help the community more reliably assess model learning potential.
>
> > **Weakness 4: [Findings 4] It’s possible that in most cases the regression models are not statistically significant, therefore unable to support the authors’ claim.**
>
> **Answer:** Thank you for your valuable comments. Our intention in calculating the slope was to evaluate the learning ability of the models. You are absolutely right—when the sample size is insufficient, regression fitting may not be significant, which can introduce bias. Therefore, we have chosen to use the quarter-on-quarter growth rate (QoQ) instead of the fitted slope, as QoQ can more accurately reflect the model’s improvement at different positions. As with other metrics, we conduct **five** runs and report the mean values to ensure reliability, as shown in the table for Weakness 3. From the experiments, we observed that Qwen and GPT-4o exhibit higher QoQ values, likely due to lower initial accuracy and greater room for improvement. Claude-3.7-Sonnet-Thinking, with higher initial accuracy, also achieves a comparable QoQ, suggesting strong learning ability. Some models have QoQ values close to zero, possibly due to already high starting points or weaker learning abilities.
>
> We will carefully revise our manuscript to include the new metric and updated results, providing readers with a clearer understanding of the models’ learning behaviors.
>
> > **Weakness 5: [Findings 6] The authors claim that Demonstration learning yields better results than Few-shot parallel solving, but isn’t that expected since demonstration learning overall has more in-context learning examples?**
>
> **Answer:** Thank you for your valuable question， but we’d like to clarify this issue from two perspectives:
>
> (1) First, the statement you mentioned is not the **main point** of our Finding 6, but rather a supporting observation. The central message of Finding 6 is that "Different solving approaches significantly affect model performance, and learning capability is not strongly correlated with a model’s inherent static ability" (as highlighted in our manuscript).
>
> (2) Second, although it is expected that Demonstration Learning, which provides more in-context examples, would outperform Few-shot Solving, our experiments reveal counterexamples. For example, DeepSeek-V3 achieves better performance with Few-shot Solving than with Demonstration Learning on most tasks (4 out of 6). This shows that the relationship between the number of in-context examples and model performance is **not always straightforward.**
>
> In summary, our statement about Demonstration Learning vs. Few-shot Solving is necessary because it highlights an important observation: while the expected trend generally holds, there are notable counterexamples. This suggests that models **vary** in their capacity to utilize demonstrations for task solving. Moreover, this is a statement mentioned in Finding 6 and does not represent the main point of Finding 6. Instead, Finding 6 emphasizes that a model’s inherent static ability does not fully determine its dynamic learning potential as evaluated by EvaLearn, which is an important but underexplored aspect.
>
> > **Question 1: How do you prompt the judge model to give feedback for Feedback learning? In the Appendix, only the prompt for grading is provided.**
>
> **Answer:** The prompt for the judge model instructs it to output both a grading rationale and a score in the specified format. The grading rationale provides feedback on the correctness of the student model’s response, as illustrated in Figure 19. The prompt used for Feedback Learning is shown in Figure 23, which provides the student model with historical experience.
> We will add a brief description of the judge model’s feedback process to Section 3.1 (Setup) of our latest manuscript to make our paper clearer.
>
> > **Question 2: How did you construct the judge rubrics for the summarization task? Unlike the rest of the task, summarization is relatively subjective.**
>
> **Answer:** We would like to clarify that the manuscript includes a description of the process of constructing rubrics for problems with multiple correct answers, including those in summarization tasks; however, **due to space limitations**, this content is presented in Appendix C.
>
> In brief, for problems that allow multiple correct answers or involve more open-ended tasks such as summarization, annotators create rubrics that either comprehensively cover all acceptable solutions or develop detailed, general rubrics that specify the required conditions for correctness.
>
> For example, in a summarization task, the rubric may list all key information from the original text and require the judge to systematically verify that each key point is correctly included in the generated summary. However, some extremely open-ended problems are difficult to evaluate objectively (e.g., certain brainstorming questions). Annotators exclude such problems to ensure the accuracy and reliability of rubric-based evaluation. Furthermore, each rubric undergoes strict validation in real judging scenarios, as detailed in Appendix C.
>
> > **Question 3: Clarification of “Sequential Reasoning” Task and Writeup/Style.**
>
> **Answer:** Sequential reasoning refers to solving problems by following a strict step-by-step order, where each step depends on the result of the previous one. This emphasizes the importance of sequence and dependency in the reasoning process. This is different from logical reasoning, which does not necessarily require strict sequential steps but instead focuses on logical analysis and the soundness of inference.
>
> We will enrich the definitions of task categories in EvaLearn to help readers better understand the composition of the benchmark. Moreover, we will carefully review our paper to correct typos and further improve its readability.
>
> Thank you again for your constructive feedback, which has greatly improved EvaLearn. We will incorporate all new experiments and revisions in the next manuscript version. If you have further questions, please feel free to contact us.

---

> ### Comment · Reviewer_xW5S · 2025-08-03
> **Response to rebuttal**
>
> Thank you for your response.
>
> I especially appreciate that you have conducted additional experiments with multiple runs to show the robustness of your results.
>
> I believe my concerns are mostly addressed, some of which were due to misunderstandings.
>
> Overall, I believe that this work is a valuable contribution. I'll revise my score accordingly.

---

> ### Author Response · Authors · 2025-08-03
> **Response to Reviewer**
>
> Thank you for your response and support! We're glad that the concerns and misunderstandings have been addressed. We truly appreciate your valuable comments once again!

---

### Official Review · Reviewer_5iax · 2025-07-04

**Clarity:** 3
**Significance:** 3
**Originality:** 4
**Rating:** 5
**Confidence:** 2

**Summary:**

The paper introduces a benchmark to evaluate a LLM's learning capability and efficiency in challenging tasks. It contains 648 challenging problems arranged into 182 seven-item sequences spanning six task types (Summarization, Classification, Extraction, Logical Reasoning, Mathematical Reasoning, Sequential Reasoning).

**Questions:**

1. Could you provide a justification on why you think 182 sequences a good number to evaluate any model. Especially when each category has about 48 or 60 problems (13 or 16 sequences) only as show in table 1. In my opinion 13 or 16 is not a significant number to evaluate sequential learning ability of the frontier models.
2. From figure 3 (right) it seems that average is between 1-2. Which means that most examples sequences are predicted correctly in the first step only. As the models get better and better we need more challenging benchmarks to evaluate these models. In the existing results the models already perform well on the dataset.
3. More data with difficultly rating as labels would have been very useful to evaluate a wide range of models with different capability and study how difficulty rating impacts it.

**Ethical Concerns:**

["NO or VERY MINOR ethics concerns only"]

**Final Justification:**

The rebuttal address all my questions. The paper is very useful and timely for the community according to me. I didn't any major issues even before the rebuttal.

**Quality:**

3

**Strengths And Weaknesses:**

Strengths:
1. Original and novel idea. Significant to the community.
2. Thorough analysis and automatic-evaluation pipeline.

Weakness:
1. I am not sure if 648 problems are enough to evaluate the sequential learning task properly.
2. To evaluate properly, there should be a range of difficultly level of these questions so that one can understand the true potential of the LLM for different difficulty levels.

---

> ### Author Rebuttal · Authors · 2025-07-30
>
> Thank you very much for providing valuable suggestions. We have addressed each of the issues you raised and will incorporate the corresponding revisions into our latest manuscript.
>
> > **Weakness 1 & Question 1: I am not sure if 648 problems are enough to evaluate the sequential learning task properly. Could you provide a justification on why you think 182 sequences a good number to evaluate any model. Especially when each category has about 48 or 60 problems (13 or 16 sequences) only as show in table 1. In my opinion 13 or 16 is not a significant number to evaluate sequential learning ability of the frontier models.**
>
>
> **Answer:** Thank you for your valuable comments. We would like to justify the number of problems in EvaLearn from the following perspectives:
>
> (1) Each problem in EvaLearn is **high-quality.** All problems in EvaLearn are annotated by human labelers and have undergone a rigorous process for data selection and quality improvement to ensure both difficulty and quality. Due to space limitations in the main text, we have included this in Appendix C. Additionally, each problem is accompanied by a human-written, instance-level rubric for automated evaluation. Therefore, although EvaLearn contains 648 problems, every problem is of high difficulty and quality.
>
> (2) The sequences in EvaLearn are thoughtfully designed. While the total number of sequences per category may be limited, each sequence is centered around a specific, related task. A complete sequence allows us to evaluate whether a model can progressively learn from its past problem-solving experiences to address new, related problems. From this perspective, the 182 sequences in EvaLearn provide a meaningful assessment of a model’s potential for sequential learning.
>
> (3) The evaluation is stability and reliability: We also followed the advice of Reviewer xW5S by conducting multiple evaluations and reporting variances (as shown in our response to Weakness 3). The results show that the confidence intervals are **narrow**, indicating stable and reliable evaluation outcomes. This further demonstrates the effectiveness of EvaLearn as an evaluation benchmark.
>
> (4) Comparison to other pioneering and popular benchmarks: EvaLearn introduces a novel evaluation dimension by being the first to assess models’ learning ability and learning potential. As a pioneering benchmark, EvaLearn aims to inspire a shift in the LLM evaluation community **from static to dynamic evaluation paradigms.** For reference, other pioneering benchmarks that evaluate specific model capabilities also contain a limited number of problems. For example, HumanEval [1] contains **164** problems to evaluate code generation of LLMs; AIME-2024 [2] has **30** problems to assess competition-level mathematical reasoning; and Paperbench [3] uses **20** machine learning-related papers to assess repository-level code reproduction.
>
> Despite the smaller number of problems, these benchmarks have become mainstream and widely adopted due to their novel evaluation focus, sparking further research and targeted model improvements. We hope EvaLearn will similarly inspire the research community to focus on models’ learning potential and stimulate future research in this direction.
>
> In summary, EvaLearn is a novel benchmark that evaluates the learning potential of SOTA LLMs. In the future, we plan to further enhance EvaLearn’s coverage by annotating more high-quality sequence data, expanding the range of categories, and extending EvaLearn to the multimodal domain to provide a more comprehensive assessment of current models’ learning potential.
>
> Thank you again for your valuable comments. We will include this statement in the latest version of our manuscript to help readers better understand our contributions and future research directions, and to inspire further work in this new area.
>
>
>
> > **Weakness 2 & Question 3: To evaluate properly, there should be a range of difficultly level of these questions so that one can understand the true potential of the LLM for different difficulty levels. More data with difficultly rating as labels would have been very useful to evaluate a wide range of models with different capability and study how difficulty rating impacts it.**
>
>
>
>
> **Answer:** Thank you for your valuable suggestions! We would like to clarify that, during the annotation process in EvaLearn, each problem was assigned to one of three coarse-grained difficulty levels, based on a combination of human judgment and the average accuracy of SOTA models. The difficulty level for each problem can be found in the **Supplementary Material** of our submission (specifically under the key "level"). Since the categorization of difficulty is relatively broad and coarse-grained, and different annotators may have varying interpretations of problem difficulty, we did not investigate the impact of difficulty on sequential learning in EvaLearn to avoid presenting unsubstantiated findings.
>
> We will include a description of the dataset difficulty levels in our latest manuscript to help users better understand the difficulty of each problem. In the future, we plan to refine the definition of problem difficulty. We also intend to explore how difficulty affects models’ sequential learning abilities, including performance on sequences with increasing difficulty and the differences in sequential learning performance between sequences of easier and harder problems.
>
>
> > **Question 2: From figure 3 (right) it seems that average is between 1-2. Which means that most examples sequences are predicted correctly in the first step only. As the models get better and better we need more challenging benchmarks to evaluate these models. The models performed well on the dataset in the existing results.**
>
> **Answer:** Thank you for your valuable suggestions. We would like to clarify the following: the right side of Figure 3 shows the performance of OpenAI-o3-mini on the  $P_\text{first}$ metric, as indicated in the figure’s caption. In fact, among all the frontier models evaluated, OpenAI-o3-mini achieves the best performance on this metric. **Due to space limitations in the main text**, other performance results can be found in Figure 11 of the Appendix. Figure 11 shows that, from the perspective of the  $P_\text{first}$ metric, sequential learning remains **highly challenging** for most LLMs except OpenAI-o3-mini. For example, in the math reasoning category, the majority of LLMs do not exhibit strong learning abilities—Claude-3.7-Sonnet, for instance, has a  $P_\text{first}$ of only 5.54.
> Moreover, we conducted additional experiments to decouple the static ability of the models from their dynamic learning efficiency within the  $P_\text{first}$ metric (as shown in the response to Reviewer djCX's Question 2). The experimental results indicate that, when focusing more on learning ability, all models show significant performance drops while exhibiting clear distinctions, demonstrating that the new metric is both challenging and effective.
>
> In summary, all problems in EvaLearn are annotated by experts and have undergone rigorous filtering for difficulty and quality. As the first benchmark to evaluate models’ learning potential, EvaLearn presents significant challenges even for the most popular and advanced LLMs. We believe that EvaLearn will encourage the academic community to explore new methods to further improve the dynamic learning abilities of LLMs. Moreover, in the future, we will continue to expand the range and difficulty of problems in EvaLearn, providing ongoing benchmarks to guide the improvement of models’ learning abilities.
>
> We will also revise the caption of Figure 3 to include a reference to Figure 11, to help readers better understand the performance of all models and improve the readability of our manuscript.
>
>
> Finally, we sincerely thank the reviewer for all the valuable suggestions provided for our manuscript. We will make sure to incorporate all revisions into the next version of our manuscript. We also plan to continue expanding EvaLearn in the future to further advance the evaluation of dynamic learning capabilities. If you have any further comments, please do let us know, and we will do our best to address them.
>
> **References:**
>
> [1] Chen M, Tworek J, Jun H, Yuan Q, Pinto HP, Kaplan J, Edwards H, Burda Y, Joseph N, Brockman G, Ray A. Evaluating large language models trained on code. arXiv preprint arXiv:2107.03374. 2021 Jul 7 (OpenAI).
>
> [2] AIME 2024 benchmark on Huggingface.
>
> [3] Starace G, Jaffe O, Sherburn D, Aung J, Chan JS, Maksin L, Dias R, Mays E, Kinsella B, Thompson W, Heidecke J. PaperBench: Evaluating AI's Ability to Replicate AI Research. arXiv preprint arXiv:2504.01848. 2025 Apr 2 (OpenAI).

---

> > ### Comment · Reviewer_5iax · 2025-08-07
> >
> > Thanks for the Nice rebuttal. I have raised my score.

---

> > > ### Author Response · Authors · 2025-08-07
> > > **Response to Reviewer**
> > >
> > > Thank you very much for your response! We are pleased that our rebuttal has addressed your questions. Thank you once again for your recognition and support of our work!

---

> ### Comment · Area_Chair_HAfq · 2025-08-05
> **Reminder: Reviewer 5iax first response**
>
> Dear reviewer 5iax,
>
> This is a reminder to post your first response, as the deadline of author-reviewer discussion period is closing. The authors have responded to your reviews, and also to the others reviews. Please discuss openly with the authors, regarding your reviews and the addressed questions from the authors.

---

### Comment · Area_Chair_HAfq · 2025-08-05
**Reminder: end of author-reviewer discussion period**

Dear reviewers,

This is a reminder that the end of author-reviewer discussion period is near. Please do carefully read all other reviews and the author responses; and discuss openly with the authors, especially on your own questions that the authors addressed.

If you have not posted your first response to the author responses, please do so now.

Thank you.

---

### Note · Authors · 2025-08-13

We express our sincere gratitude to all reviewers for their valuable feedback, active engagement, and final recognition of our work. We deeply appreciate your thoughtful evaluations, which highlight several strengths of our paper:

1. Well-motivated and novel evaluation perspective on LLM learning capability and efficiency (Reviewers 5iax, xW5S, 6LKa, djCX)
2. Thorough experimental design and high-quality benchmark, including challenging, rubric-verified problems and comprehensive automated metrics (Reviewers 5iax, 6LKa, djCX)
3. Clear presentation and reproducibility (Reviewers 6LKa, xW5S, djCX)
4. Significant contribution to the community, providing a concrete benchmark for assessing LLM learning potential (Reviewers 5iax, djCX)

Reviewer xW5S suggested reporting the statistical significance of our experiments, including significance testing and multiple runs to ensure robustness. Reviewer 6LKa recommended a cross-LLM agreement study. Some reviewers also asked about the rationale behind EvaLearn’s problem and sequence design, and raised questions about evaluation metrics. Reviewer 6LKa also suggested expanding the discussion of earlier interactive and curriculum-based work.

We have provided detailed responses to all concerns in our rebuttal, which received substantial recognition from the reviewers. Specifically, for statistical significance and reliability, we conducted additional experiments with five runs using different random sequence compositions, and reported 95% confidence intervals. The results show that our evaluation is stable and robust across different runs. For the cross-LLM agreement study, we evaluated rubric-based judging using multiple LLMs. The results show an agreement rate above 98% and Cohen’s κ values exceeding 0.96, confirming that our automated evaluation is highly reliable.

We will further refine our paper based on the rebuttal content to make our work even more solid. Moreover, we will comprehensively open-source EvaLearn, including the dataset, evaluation code, and all benchmarked model results, to significantly facilitate the advancement of LLMs’ learning capabilities.

Finally, we greatly appreciate the reviewers’ unanimous recognition of EvaLearn's distinctive contribution and its potential to advance the field of dynamic evaluation. We are confident that our work provides a foundation for future exploration of LLMs’ learning potential, and believe it will make a valuable contribution to the field.

---

### Decision · Program_Chairs · 2025-09-17

**Decision:**

Accept (poster)

**Comment:**

The paper introduces EvaLearn, a benchmark to measure within-task learning of LLMs via sequences of related problems across six task families. Models are evaluated sequentially within every sequences from 648 challenging problems arranged into 182 sequences spanning six task families, and evaluated automatically using instance-level rubrics and LLM-as-a-judge. Ultimately, the model are then evaluated using five proposed metrics of overall accuracy, slope of position-wise accuracy curve, position of first success, longest correct streak, and post-warm-up accuracy, to quantify learning capability and efficiency. The author conducted wide range of analysis, and interestingly, static leaderboard strength is only weakly correlated with sequential learning skill.

The work is novel, timely, methodologically sound, and addresses the key gap of current evaluation paradigm. Rebuttal additions convincingly resolve the main methodological concerns (statistics, insights, metric robustness, protocol clarity, judge reliability). Overall, I believe the contribution of this work would be meaningful and interesting to the community.

__Strengths__
* Evaluation shift from static competence to within-task learning, is novel and timely
* Benchmark is high-quality and evaluation methods are meaningful
* Analysis are thorough
* Paper is well-written and organized

__Weaknesses__
- The rebuttals have addressed most—if not all—of the weaknesses and concerns.